# Frame In-N-Out: Unbounded Controllable Image-to-Video Generation

**Boyang Wang**[1]  **Xuweiyi Chen**[1]  **Matheus Gadelha**[2]  **Zezhou Cheng**[1]

[1]University of Virginia  [2]Adobe Research

**Project Page:** `https://uva-computer-vision-lab.github.io/Frame-In-N-Out/`

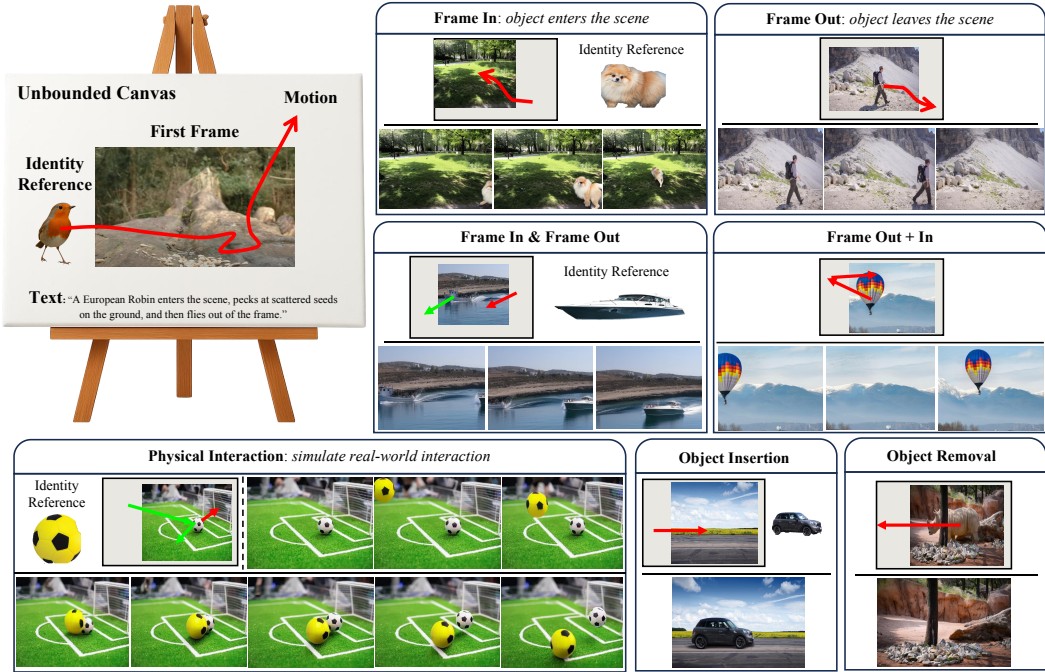

Figure 1: **Frame In-N-Out** presents a new task in the image-to-video generation that extends the first frame into an *unbounded* canvas region, where the model could be conditioned on identity reference with motion trajectory control to achieve Frame In and Frame Out cinematic technique.

## Abstract

Controllability, temporal coherence, and detail synthesis remain the most critical challenges in video generation. In this paper, we focus on a commonly used yet underexplored cinematic technique known as Frame In and Frame Out. Specifically, starting from image-to-video generation, users can control the objects in the image to naturally leave the scene or provide breaking new identity references to enter the scene, guided by a user-specified motion trajectory. To support this task, we introduce a new dataset that is curated semi-automatically, an efficient identity-preserving motion-controllable video Diffusion Transformer architecture, and a comprehensive evaluation protocol targeting this task. Our evaluation shows that our proposed approach significantly outperforms existing baselines.

## 1   Introduction

While watching a movie, the frame presented to the viewer only shows a deliberately chosen portion of the scene. The director's imagination extends far beyond what is shown on screen. Crucial plot twists

arise outside the frame that the viewer can observe. For instance, a director may decide to introduce a new character into the scene to enhance dramatic tension, or conversely, ask a character to exit the frame for subsequent plot progression. As modern video generation advances toward producing more controllable and high-fidelity content, a natural question arises: can we enable video generation to capture a wider and more imaginative world that is not confined by the spatial boundaries of the initial frame? In this paper, we are targeting achieving such a milestone in controllable video generation by making real-world cinematic techniques of **Frame In** and **Frame Out** (Frame In-N-Out) come true.

We formalize Frame In and Frame Out as an image-to-video generation setting. Starting from the first frame image (as shown in Fig. 1), our goal is to create a model that, given an explicit motion trajectory, can (i) control an existing object in the frame completely outside the visible bounds of the first frame and subsequently bring it back while preserving fidelity and integrity, which is defined as Frame Out, and (ii) allow a new identity (ID) object (e.g., humans, vehicles, animals) to enter the scene plausibly—whether from the sides or from above, which is defined as Frame In. Contemporary motion-controlled image-to-video generation architecture design like [81, 14, 61, 87, 77, 44, 52] needs spatiotemporal pixel-aligned trajectory signal to the first frame, treating the image border as an immovable "wall", which we can see from Fig. 4. To transcend this border-bounded constraint, we extend the conditioning control beyond the first frame region size to an **unbounded canvas region**. Conditioning signals are applied over this enlarged canvas, enabling the object to move out of the first frame region or using the canvas region to prepare the entrance of a new identity (ID) reference while maintaining temporal and spatial coherence.

The overall design of the Frame In-N-Out framework is under study. A key challenge lies in the absence of existing training datasets that explicitly capture Frame In and Frame Out dynamics. To address this, we redesign the data curation pipeline from scratch. This includes identifying track-worthy objects, improving tracking reliability, and defining suitable bounding boxes that partition the first frame from the extended canvas region. A new video Diffusion Transformer [76, 40] is needed to integrate multiple conditions that are either spatiotemporal pixel-aligned [32, 77, 67] (*e.g.*, motion), unaligned [33, 12] (*e.g.*, identity), and more importantly, unbounded canvas. As Frame In-N-Out represents a new task in the video generation domain, we meticulously curated a dedicated evaluation system. This includes constructing benchmark datasets for both Frame In and Frame Out scenarios and revising traditional tracking, and identity-preserving metrics to reflect the unique demands of this problem setting. We believe this formulation can inspire future research into more expressive and application-driven conditioning research in video generation.

In summary, this work makes the following contributions:

- To the best of our knowledge, this is the first attempt to explore Frame In and Frame Out patterns in video generation.
- We define and curate a training dataset for the Frame In and Frame Out pattern recognition. The pre-processing code and metadata will be released.
- We propose an efficient controllable video Diffusion Transformer that unifies spatiotemporal pixel-aligned motion, pixel-unaligned identity reference, and our proposed unbounded canvas conditions in one model.
- We provide an evaluation system for the Frame In and Frame Out scenarios and showcase that ours outperform other baselines. Our insight and area focus have broad prospects in scenarios such as the film industry or advertising production.

## 2   Related Works

**Base Condition in Video Generation.** While numerous conditioning strategies have been proposed for video generation, some base conditioning exerts a fundamental influence on the eventual generation quality and choice of the base model. Broadly, these can be categorized into three primary paradigms: Text-to-Video [5, 17, 9, 56], Image-to-Video [7, 76, 56], and Video-to-Video [31, 75, 37, 53, 3, 85]. Vanilla text-to-video task involves generation from sparse conditioning signals, where no pixel-level guidance is needed. The model must rely entirely on language prompts to imagine and synthesize all visual content, including scene layout, motion, and object appearance. In contrast, Image-to-Video assumes the presence of a single reference frame as the first frame, from which the entire video must follow. This requires all subsequent frames to remain aligned with the initial spatial content, even if additional conditions like text prompts are provided.

Table 1: **Conditioning comparisons to existing controllable video generation works.**

| Methods | Text | First Frame | Identity | Motion | Unbounded Canvas |
|---|---|---|---|---|---|
| CogVideoX [76] | ✓ | ✓ | ✗ | ✗ | ✗ |
| MotionCtrl [67] | ✓ | ✗ | ✗ | ✓ | ✗ |
| DragAnything [69] | ✗ | ✓ | ✗ | ✓ | ✗ |
| Image Conductor [32] | ✓ | ✓ | ✗ | ✓ | ✗ |
| ToRA [87] | ✓ | ✓ | ✗ | ✓ | ✗ |
| ConsisID [79] | ✓ | ✗ | ✓ | ✗ | ✗ |
| SkyReels-A2 [12] | ✓ | ✓ | ✓ | ✗ | ✗ |
| Phantom [35] | ✓ | ✓ | ✓ | ✗ | ✗ |
| Ours | ✓ | ✓ | ✓ | ✓ | ✓ |

**Controllable Video Generation.** Controllable video generation refers to the task of extending pre-trained video generation models by modifying their architecture to incorporate one or more additional conditions beyond the original text, image, or video inputs. Since the appearance of the image diffusion models, researchers have explored a wide range of conditioning signals. This including sketches [71], human pose [27], low-quality images [68] for restoration, masked images for inpainting [24, 91], outpainting [84, 10], and editing [78, 23]. In the video generation domain, temporal-oriented challenges include interpolation between the first and last frame [22, 57], motion control [67, 72, 74, 77, 89, 66, 61], camera control [70, 67, 16, 66, 3, 72], and long-range history-guided generation [45, 80]. Further, as a flexible condition, identity (ID) reference is also broadly studied in both image and video side, like PhotoMaker [33], InstantID [65], ConsisID [79], and Concat-ID [90]. Building upon this foundation, there has been a growing trend toward elements-to-video generation, where not only identity reference images but also the first frame can serve as an individual compositional element, like Phantom [35], and SkyReels-A2 [12]. The conditioning comparisons can be found in Tab. 1.

## 3 Problem Definition

This paper focuses on solving unbounded controllable image-to-video generation. Specifically, we concretize the problem into a specific task in the cinematic domain: **Frame In** and **Frame Out**. Our controllable video generation targets at the intersection of four control signals: (1) first frame image $I_0$ and text prompt $\mathbf{y}$ as the foundation condition, (2) a canvas area expansion bounding box setting $\mathbf{B}_{canvas}$ that is composed of top-left and bottom-right pixels expansion amount, (3) motion trajectory $\mathbf{c}_{trajs}$ for an existing object in the first frame or a new identity (ID) to introduce, and (4) an optional identity reference image $\mathbf{f}$ (*e.g.*, a human, vehicle, animal, balloon, etc.). In the Frame Out case, we don't need an ID reference provided, but it is mandatory in the Frame In case. Eventually, our video Diffusion Model generates $N$ number of latent frames that strictly follow all conditions, aiming to learn a conditional joint distribution $p_\theta(I_1, ..., I_N | I_0, \mathbf{y}, \mathbf{B}_{canvas}, \mathbf{c}_{trajs}, \mathbf{f})$.

## 4 Dataset Curation and In-N-Out Pattern Recognition

Data curation and pattern recognition play a pivotal role in achieving the Frame In and Frame Out intention we want. Our curation starts with raw videos without utilizing metadata provided by the original dataset. The target is to provide an explicit, high-quality identity (ID) reference image, a clear and accurate motion trajectory, and a bounding box to partition the canvas and the first frame region. Hence, we modify the traditional curation in image-to-video generation and our curation logic is composed of the following four parts (also shown in Fig. 2). Specific hyperparameters and more setting details are in the supplementary.

**Basic Curation.** Our basic curation consists of the following steps. (1) *Metadata filtering*: we first selected videos based on the metadata attributes such as duration, resolution, and aspect ratio. (2) *Image-level filtering*: for each video, we randomly sample two frames and filter out low-quality videos using automated image quality assessment [48] and aesthetic assessment [49]. We additionally apply image complexity assessment [13] to exclude both overly simplistic and excessively complex videos, which are known to hinder learning [59]. Videos with excessive overlaid text are also filtered using an OCR detector [2]. (3) *Video-level filtering*: we remove multi-scene videos using scene cut detection from TransNet V2 [46], and discard videos with significant camera motion (*e.g.*, rotation, translation, or focal changes) based on motion estimation from CUT3R [64], focusing on

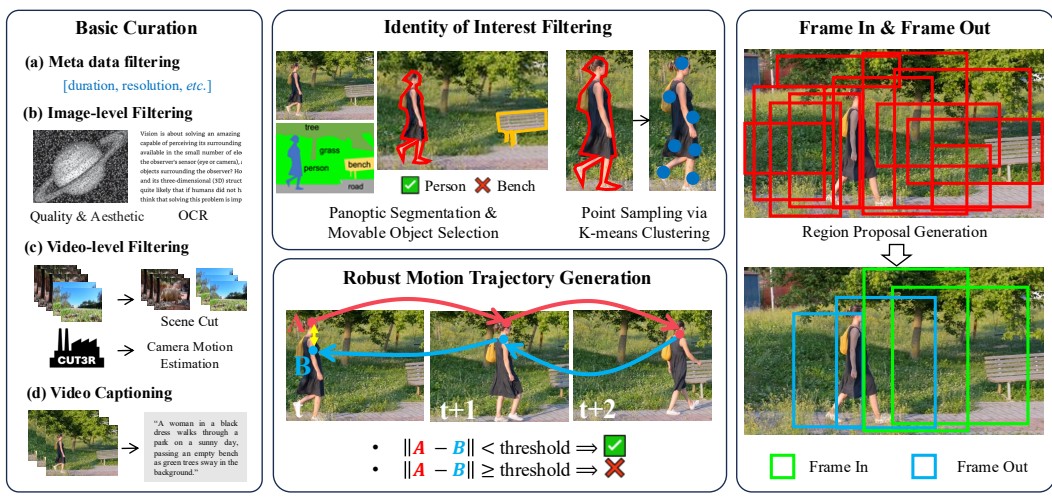

Figure 2: **Data Curation Pipeline.** Our curation pipeline will provide high-quality filtered videos, text prompts, tracking trajectories with semantic labels, and bounding boxes that can be ideal partitions between the first frame and canvas region.

object-centric motion with least impact from camera movement. (4) *Automatic Captioning*: to obtain high-quality paired text prompts, we discard dataset-provided captions and generate new ones by QWen2.5-32B-VL-Instruct [73].

**Identity of Interest.** Random point-based tracking, like optical flow [67, 87], does not provide semantic meaning to each point; i.e. each point cannot strictly correspond to an identity. Thus, before applying the tracking model, we apply panoptic segmentation [29] with OneFormer [21] to classify and then segment all objects in the image. We observe that videos with ideal Frame In and Frame Out patterns usually come with multiple relevant start frames across a video. Thus, we select 3 starter frames at the duration 0%, 35%, and 70% of the full video length. This strategy alleviates the dataset scarcity in the later stage. To get the clearest and the least compressed frames available, we apply I-Frame extraction from traditional video compression [43, 59] and choose the closest I-Frame as the official starter frame. These 3 starter frames will be taken as the first frame for image-to-video generation and also execute the panoptic segmentation. Panoptic Segmentation from OneFormer will classify 133 classes based on the COCO dataset [34]. We manually define 22 classes of them as motionable objects that could be objects of interest in the following tracking annotation. Our purpose is to filter out static objects, like trees, houses, and sightseeing, which are not ideal as a tracking target. Meanwhile, based on the size of segmentation masks, we filter both small and over-sized identity objects. Further, inspired by [61], we apply K-means to get an even distribution of points from the segmentation mask.

**Cycle Tracking:** With objects of interest and even distributed points, we apply tracking from CoTracker3 [26]. However, tracking can be unstable and inaccurate, especially on fast-moving objects. To the best of our capability to provide the most accurate tracking trajectories without human correction, we take advantage of back-tracking functionality from CoTracker3. After the regular forward track, we back-track from the end position of points in the last frame to the first frame. If the error between the initial position and the back-tracked position is larger than a preset threshold, we filter out those points. Fewer but accurate points are more helpful for motion-controllable video generation training. In the end, we sort and filter small and extremely high-motion cases to avoid static objects and over-fast movements.

**Frame In and Out Pattern:** Given a well-defined object of interest along with its trajectory information, we aim to search for bounding boxes that partition the video frame into two regions: an in-box region, as the first frame in training, and an out-of-box canvas region, which serves as the creative area for ID and unbounded motion. We adopt a regression-based strategy by randomly generating thousands of bounding boxes with varying sizes and using tracking information to identify Frame In and Frame Out patterns. To ensure sufficient diversity in the training and considering mobile screen aspect ratios, we sample boxes with various aspect ratios ranging from 16:9 to 4:5. To prevent over-small cases, each bounding box is constrained to have a height of at least 50% of the full canvas.

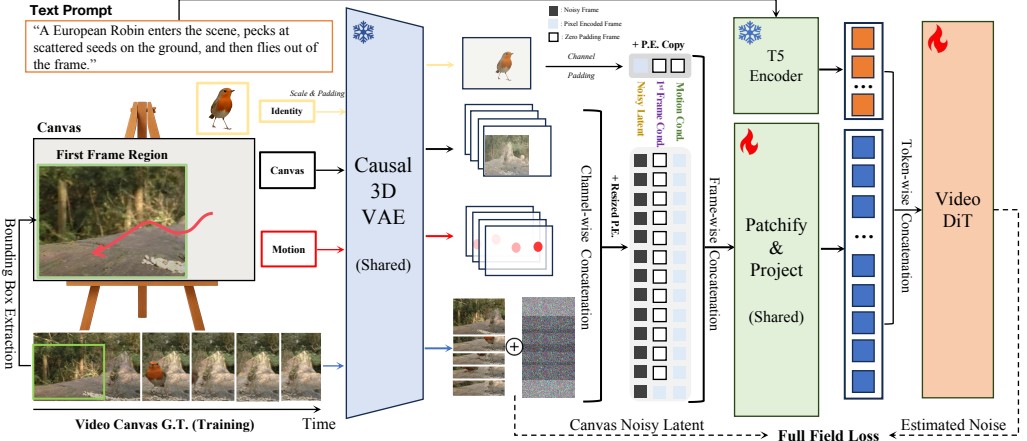

Figure 3: **Main Architecture.** Our video Diffusion Transformer embraces the first frame with canvas expansion, motion trajectories, identity reference, and text prompt as conditions for video generation.

Frame Out cases are instances where the object is initially partially or completely located inside the bounding box and subsequently moves entirely outside the bounding box in at least one frame of the video. Re-entry into the box is allowed for robust training and diverse user cases. In contrast, Frame In cases require the object to be completely outside the bounding box in the first frame, with no pixel overlap. This ensures that the ID reference has no overlap with the first frame in the training and we could condition breaking new information to the image-to-video generation. Next, in the following frames, a sufficient fraction of the object must come to the in-box region to be considered as a qualified Frame In scenario. For valid Frame In cases, we employ SAM2 [42] to extract the object mask and store the corresponding cropped ID reference image. The SAM2 mask is also used to further filter out inaccurate tracking points.

## 5    Frame In-N-Out Architecture

### 5.1    Base Architecture with Flexible Resolution Training

The base video Diffusion Transformer architecture we consider is CogVideoX-I2V [76], a relatively small-scale model (5 billion parameters) compared to tens of billions of parameter models [57, 30, 36, 15]. We believe that our contribution is scalable to most video Diffusion Transformer considering the similarity in the architecture [40]. The first frame $I_0$ is conditioned by doing channel-wise concatenation with the noisy latent $\mathbf{Z}$ before entering the transformer blocks:

$$\mathbf{Z}' = \text{Concat}_c(\mathbf{Z}, \mathcal{E}(I_0)), \tag{1}$$

where $\text{Concat}_c(\cdot, \cdot, \cdots)$ denotes concatenation along the channel dimension, and $\mathcal{E}$ is the VAE encoder. $\mathbf{I}_0$ is filled in with zero padding $\emptyset$ on the frame dimension as a placeholder for frame-wise alignment.

The default resolution supported is 480×720. In order to support different resolution in training and inference based on the needs, we take advantage of the nature of the absolute position embedding and rotary position embedding [47] (RoPE). For absolute position embedding, inspired by [1], we apply a trilinear interpolation on the learned fixed absolute embeddings to the target latent size to adapt different resolution inputs. For RoPE, we inject the current resolution grid size to create a new position embedding each time.

### 5.2    Motion Control

We first convert all spatiotemporal trajectory coordinates into a pixel marking in image forms $\mathbf{c}_{trajs}$. Since we use panoptic segmentation in the dataset curation, we have rich and accurate semantic meaning for each tracking point. Hence, we provide different objects with different color markings to promote the model learning semantic relationship. For the same object with multiple tracking points, they share the same color. Our trajectory point represents the spatiotemporal pixel-aligned state of the

---

[1]https://github.com/aigc-apps/VideoX-Fun

object which is different from optical flow-based motion vector representation as in previous motion control works [67, 58, 87].

The motion conditioning has various solutions in the literature. ControlNet-like [81, 14, 61], cross-attention based [87, 77], or extra training tokens [25] methods are all computational expensive for motion conditioning. For pixel-aligned conditions, we prefer to apply a direct, efficient, and natural solution, which is by channel-wise concatenating VAE-encoded motion images. To be specific, we encode motion images like regular RGB images by the pre-trained 3D VAE encoder $\mathcal{E}$; thus, the latent size of the motion is perfectly aligned with the first frame latent and noisy latent $\mathbf{Z}$. Then, we do channel-wise concatenation to the motion latent as the following:

$$\mathbf{Z}' = \text{Concat}_c(\mathbf{Z}, \mathcal{E}(I_0), \mathcal{E}(\mathbf{c}_{trajs})). \tag{2}$$

However, channel-wise concatenation increases the input channel number for the first projector of the Diffusion Transformer. Hereby, inspired by Marigold [28], we first zero-initialized the projector with the new input channel number setting and then filled in the overlap weights with the pre-trained weights available to decrease the training gap.

### 5.3 Unbounded Conditioning

Starting with the first frame condition $I_0$, we want the pixel-aligned motion intention to get over the border constraints. Hereby, we first extrapolate the first frame region to a larger area called the unbounded canvas region as shown in Fig. 3, which is defined by the provided top-left and bottom-right expansion pixels quantity $\mathbf{B}_{canvas}$. We define the expansion transformation from the original size of the first frame region to the canvas region coverage as $\tau_{canvas}(\cdot)$. The first frame is transformed by zero-padded $\tau_{canvas}(I_0)$. Then, we adjust the absolute and relative position encoding system by setting the top-left of the canvas region as the (0,0,0) index for temporal, horizontal, and vertical directions. Under this setting, our motion control signal can be expanded to any area inside the canvas region, which is also defined as $\mathbf{c}_{trajs}$.

**Full Field Loss.** We initially hypothesized that the generative objective should be aligned solely with the viewer-visible region, which is the first frame region, rather than the full canvas region. Accordingly, target latent $\mathbf{Z}$ is zero-padded for the region outside the first frame. However, this formulation failed to yield stable results. Motivated by the experimental observation and outpainting task, we reformulated the objective to be the full field of the canvas denoted as $\tau_{canvas}(\mathbf{Z})$. The complete formula can be expressed as:

$$\mathbf{Z}' = \text{Concat}_c(\tau_{canvas}(\mathbf{Z}), \mathcal{E}(\tau_{canvas}(I_0)), \mathcal{E}(\mathbf{c}_{trajs})). \tag{3}$$

In the training, to accommodate faster training [63] without utilizing an attention mask for different resolutions between batches [11], we resize all videos to the same canvas resolution, but the unmasked first frame region can be versatile based on the dataset curation strategy from Sec. 4. Thanks to the flexible nature of the Diffusion Transformer with absolute and relative position embedding, we can set the canvas size arbitrarily in inference with desirable results, even if we train with a fixed size.

### 5.4 Unifying Identity Reference Conditions

Our model leverages the causal 3D VAE nature in modern video Diffusion Transformer [76], where not only sets of frames can be compressed, but one single image can be encoded. Therefore, we use the same pre-trained VAE $\mathcal{E}$ to encode the Identity (ID) reference $\mathbf{f}$. Inspired by Concat-ID [90] and recent progress in text-to-image generation with ID reference [50, 86], we resize and scale the ID reference to the same resolution as the canvas size $\mathbf{B}_{canvas}$ and then do frame-wise concatenation between the latent ID reference and the video frames. The ID reference will be added with random augment noise $n$ before being encoded by VAE. To align the channel number, we add zero padding $\emptyset$ on the corresponding first frame and motion condition channels. The formula can be expressed as:

$$\mathbf{F}_{ID} = \text{Concat}_c(\mathcal{E}(\mathbf{f} + n), \emptyset, \emptyset), \tag{4}$$

$$\mathbf{F}_{Video} = \text{Concat}_c(\tau_{canvas}(\mathbf{Z}), \mathcal{E}(\tau_{canvas}(I_0)), \mathcal{E}(\mathbf{c}_{trajs})), \tag{5}$$

$$\mathbf{Z}' = \text{Concat}_f \{\mathbf{F}_{Video}, \mathbf{F}_{ID}\}, \tag{6}$$

where $\text{Concat}_f \{\cdot, \cdot, \cdots\}$ denotes frame-wise concatenation. After forwarding the Diffusion Transformer, text tokens and ID tokens will be discarded, without contributing to the loss.

This method leverages the 3D full attention nature of the video Diffusion Transformer. The text tokens, video tokens, and ID reference tokens will be token-wise concatenated after the patchification procedure and then jointly optimized together. Further, by reusing all well-trained normalization, projection, and feedforward modules, the training becomes more stable and the implementation becomes more elegant. Though OmniControl [50] and EasyControl [86] might apply a shifted offset for the position encoding, this method does not provide a similar data distribution for the learned fixed position encoding on the ID reference part in the model like CogVideoX [76]. We empirically find that directly copying the position encoding of the first frame to the ID reference frame leads to better numerical results.

## 5.5 Training

Our training is composed of two stages. In the first stage, we include motion control based on the Eq. 2 with the text prompt captioned from [73] to learn the fundamental conditioning and adapt to the absolute position encoding modification we have done. Our loss in this stage can be formulated as:

$$\mathcal{L} = \mathbb{E}_{\mathbf{z}, \epsilon \sim \mathcal{N}(\mathbf{0}, \mathbf{I}), t, \mathbf{c}} \left[ ||\epsilon - \epsilon_\theta(\mathbf{z}_t, y, I_0, \mathbf{c}_{trajs}, t)||_2^2 \right], \tag{7}$$

where $t$ is the timestep, and $\mathbf{z}_t$ is the noisy latent at timestep $t$. During the inference, pure noise $\mathbf{z}_T$ is gradually denoised from timestep $T$ until timestep 0 to a clean latent $\mathbf{z}_0$. Then, it will be decoded by the pre-trained VAE decoder $\mathbb{D}$ to convert back to pixel space.

In the second stage, we jointly train Frame In and Frame Out with unbounded canvas region setting together based on Eq. 6. We consider at most one ID reference $\mathbf{f}$ each time. If it is a Frame Out only case, we insert a monocular white color placeholder $\emptyset$ on the ID reference $\mathbf{f}$ position in Eq. 6. The loss in this stage is:

$$\mathcal{L} = \mathbb{E}_{\mathbf{z}, \epsilon \sim \mathcal{N}(\mathbf{0}, \mathbf{I}), t, \mathbf{c}} \left[ ||\epsilon - \epsilon_\theta(\tau_{canvas}(\mathbf{z}_t), y, \tau_{canvas}(I_0), \mathbf{c}_{trajs}, \mathbf{f}, t)||_2^2 \right], \tag{8}$$

We observe that perfect Frame Out cases with complete move-out are rare. For Frame In, it is even harder to find cases in which ID completely has no overlap with the first frame region. To solve data scarcity for generalized and robust training, we lower the standard in the training dataset curation, where we do not require the object to be completely out or inside the first frame region. We believe that the hardest training objective is learning new ID reference signals with motion control in the original video Diffusion Transformer. The overall model structure can be found in Fig. 3. The inference pipeline can be found in the supplementary.

## 6 Experiment

### 6.1 Implementation Details

The training dataset we use includes OpenVid-1M [38], VidGen-1M [51], and subset of Webvid-10M [4]. The specific data and filtering statistics can be found in the appendix. We use the reserved subset of the OpenVid-1M dataset as our evaluation test set for Frame In-N-Out curation. Our training is on a total batch size of 8 for 32K and 50K iterations in two stages, respectively. The training resolution, which is also the canvas resolution, is 384×480 for two stages. All the video is curated, processed, and fetched at 12 FPS standards. We apply the learning rate warmup for each stage of training in the first 400 steps. The learning rate is 2e-5. Our inference step is 50 with classifier-free guidance [19]. The first frame and text dropout ratio is 5% each in the training to augment the classifier-free guidance in the inference. To make the motion pattern in the dataset stronger, we randomly doubled the duration fetched to simulate a speed-up. We randomly drop the ID reference with a probability of 15% in the stage 2 training, where we only have Frame Out to consider.

### 6.2 Proposed Evaluation Dataset Overview

Since we are the first work focusing on the In-N-Out pattern in video generation, we define evaluation datasets and metrics as follows. Our evaluation is composed of two parts: Frame Out and Frame In with identity (ID) reference. Though we don't require perfect Frame In and Frame Out patterns in

our training scenarios, for the expression of a fair intention, we set the setting to the hardest level in the curation of the evaluation test set. In this way, we curate 183 and 189 cases for ideal FrameIn and FrameOut as evaluation datasets. All Frame In evaluation datasets will come with one and only one ID reference image. The benchmark will be released for future study.

## 6.3 Evaluation Metrics

We evaluate the generative quality of video models using three widely adopted metrics: Fréchet Inception Distance [18] (FID), Fréchet Video Distance [54] (FVD), and Learned Perceptual Image Patch Similarity [82] (LPIPS). Beyond these generative metrics, we modify traditional tracking, segmentation, LLM evaluation, and identity-preserving metrics to fit the In-N-Out pattern.

**Trajectory Error** (Traj. Err.) evaluates the Euclidean distance of all trajectory points between the GT and the generated videos estimated by the Co-Tracker3 [26]. Different from trajectory error metrics proposed in [67, 69], our In-N-Out scenario considers the full canvas region for both GT and generated videos. Since the baseline method cannot generate pixels out of the first frame, they will be zero-padded to the same resolution as the GT, which refers to the canvas size. Lower scores indicate more aligned motion controllability. This metric is intended to leverage the low-level accuracy of the tracking when the object leaves and re-enters the scene.

**Video Segmentation Mean Absolute Error** (VSeg MAE) can be formulated as:

$$\text{VSeg MAE} = \frac{1}{F \times H \times W} \sum_{i=1}^{F} \sum_{j=1}^{H} \sum_{j=k}^{W} |M_{gen}(i, j, k) - M_{gt}(i, j, k)|, \tag{9}$$

where $F$, $H$, and $W$ refer to the frame number, video height, and width. $M_{gen}$ and $M_{gt}$ refer to the segmentation area of the generated video and the Ground-Truth video estimated by SAM2 [42]. The tracking points estimated by Co-Tracker3 served as the visual prompt for segmentation. Generated videos by the baseline methods without expansion capability will also be zero-padded. This metric is intended to calculate the accuracy of Frame In and Frame Out from a high-level semantic perspective.

**Vision Language Model evaluation** (VLM) utilizes a SOTA open-source vision language model, Qwen 2.5 VL 32B [73] to justify if there is any object gets out of the first frame or enters the first frame. The Frame In instruction prompt is *Please check if the object enter the frame. Return a Yes/No as the only response.* The Frame Out instruction prompt is *Please check if the object leave the frame. Return a Yes/No as the only response.* We evaluate the ratio of correctness compared to the returns with GT video inputs. If the reponse is not *yes* or *no*, we will skip that cases; however, we do not observe response different from these two designated outputs, which is thanks to strong capability of Qwen2.5. Due to computation concern, we evenly sample 14 frames from the full video sequence. The video is based on non-padded results, which does not consider the canvas region. This metric is intended to align the overall subjective success rate analysis, and the higher the better.

**Relative DINO** (Rel. DINO) inherits the traditional DINOV2 [39] from VBench [20, 88] by calculating the cosine similarity between the ID reference and each video frame for Frame In comparison. However, we found that in our Frame In-N-Out setting, there exists numerous frames that the ID reference does not appear in the video frames at all, which leads to a very low similarity score. Thus, we first calculate the average DINO similarity score between ID reference and each frame and then focus on the absolute relative difference to the Ground-Truth DINO results:

$$\text{Relative DINO} = \frac{\left| \frac{1}{F} \sum_{t=1}^{F} \langle d_{ID} \cdot d_t^{GEN} \rangle - \frac{1}{F} \sum_{t=1}^{F} \langle d_{ID} \cdot d_t^{GT} \rangle \right|}{\frac{1}{F} \sum_{t=1}^{F} \langle d_{ID} \cdot d_t^{GT} \rangle}, \tag{10}$$

where $\langle \cdot \rangle$ is the dot product operation for calculating cosine similarity. The video is based on non-padded results, which does not consider the canvas region. The lower the score, the better.

## 6.4 Frame Out Comparisons

We consider SOTA motion controllable image-to-video (I2V) model, including DragAnything [69], Image Conductor [32], and ToRA [87]. For these baselines, their architecture does not support conditions of motion trajectory points outside the first frame; thus, we implement these points not appear in the conditioning motion images. By default, all motions only apply to one point and

Table 2: **Frame Out Comparison with Motion Controllable Models**. The best is highlighted.

| Method | FID↓ | FVD↓ | LPIPS↓ | Traj Err.↓ | VSeg MAE↓ | VLM ↑ |
|---|---|---|---|---|---|---|
| DragAnything [69] | 48.73 | 607.44 | 0.462 | 41.24 | 0.0480 | 0.624 |
| Image Conductor [32] | 99.29 | 1154.86 | 0.528 | 42.72 | 0.0552 | 0.544 |
| ToRA [87] | 57.83 | 566.78 | 0.362 | 40.72 | 0.0750 | 0.603 |
| **Ours** (Stage1 Motion TI2V) | 38.36 | 478.96 | 0.358 | 48.46 | 0.0572 | 0.685 |
| **Ours** (Stage2) | **32.02** | **318.38** | **0.268** | **17.85** | **0.0229** | **0.735** |

Table 3: **Frame In Comparison with Elements-to-Video Models**. Tracking and Segmentation for the elements-to-video models are omitted because of the failure to identify the object's position in the generated videos. The best is highlighted.

| Method | FID↓ | FVD↓ | LPIPS↓ | Traj. Err.↓ | VSeg MAE↓ | Rel. DINO↓ | VLM↑ |
|---|---|---|---|---|---|---|---|
| SkyReels-A2 [12] | 74.00 | 655.25 | 0.604 | – | – | 3.37 | 0.448 |
| + Motion Description Prompt | 61.20 | 550.26 | 0.564 | – | – | 2.01 | 0.535 |
| Phantom [35] | 69.55 | 742.15 | 0.571 | – | – | 1.70 | 0.415 |
| + Motion Description Prompt | 72.84 | 671.05 | 0.596 | – | – | **1.39** | 0.540 |
| **Ours** (Stage2) | **30.84** | **227.30** | **0.218** | **10.37** | **0.0112** | 1.62 | **0.863** |

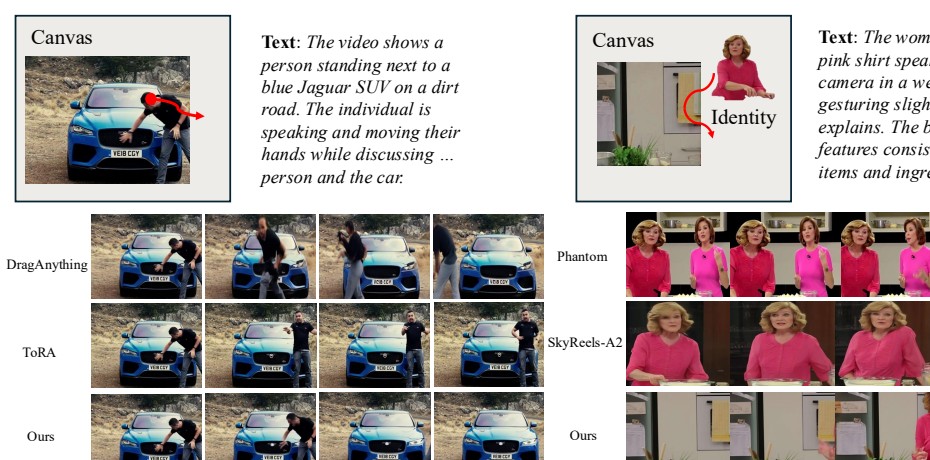

Figure 4: **Qualitative comparison on our benchmark dataset.** In (a), we compare our model on Frame Out cases against DragAnything [69] and ToRA [87]. Both baselines fail to fully move the person outside the image boundaries, while our model successfully handles a complete exit. In (b), we evaluate Frame In scenarios against Phantom [35] and SkyReels-A2 [12]. Only our model can reach Frame In effect with the designated identity.

one object for a fair comparison. Further, we include our stage 1 motion-guided image-to-video generation results.

Tab. 2 reports results across multiple metrics. Our Stage 1 model already outperforms prior methods in terms of perceptual quality (LPIPS), temporal consistency (FVD), and LLM automatic evaluations (VLM). With Stage 2 refinement, our method achieves the best results across all metrics, reducing trajectory error by over 50% compared to Drag Anything and halving the VSeg MAE. These results demonstrate the effectiveness of our Frame Out architecture advantages and also highlight the effectiveness of two-stage training.

## 6.5 Frame In Comparisons

We define the task as a conditioning generation that requires an identity (ID) reference image with the first frame and motion trajectory. This task is not well-defined before this paper. Thus, there does not exist an appropriate strong baseline in the literature. Based on the conditioning we need, we found that the recent rising elements-to-video generation (E2V) is the closest fit, which can take conditions of ID reference and the first frame. While some other text-to-video generation works like Direct-A-Video [74] have motion control with ID reference, these works are not conditioned on

the first frame, so we do not believe that they are a good fit to express the unbounded controllable video generation concept we want to present in this paper. For the E2V, we consider Phantom [35] and SkyReels-A2 [12], which is based on Wan2.1 [57] Diffusion Transformer. Further, for a fair comparison, we re-generate a more motion descriptive text prompt by LLM [73] to compensate that E2V models cannot take in motion conditioning.

As shown in Tab. 3, our method significantly outperforms prior elements-to-video models across all key metrics, including FID, FVD, LPIPS, Traj. Err., VSeg MAE, and VLM accuracy, demonstrating superior visual quality and precise motion controllability in the Frame In setting. While our Rel. DINO score is slightly less (-0.23) than Phantom with motion prompts; this is primarily due to our model faithfully following motion guidance that occasionally moves the identity outside the canvas, affecting frame-wise similarity.

## 7 Conclusion

In the paper, we have presented Frame In-N-Out, a new paradigm in image-to-video generation that gets over the border-boundary constraints from the first frame. We leverage text, motion, identity reference, and this new unbounded canvas concept to promote video generation to be more controllable and aligned with real-world applications. Our experiments demonstrate that our generated videos align with the condition intention we introduce and can foresee a border impact.

## 8 Acknowledgement

The authors acknowledge the Adobe Research Gift, the University of Virginia Research Computing and Data Analytics Center, Advanced Micro Devices AI and HPC Cluster Program, Advanced Cyberinfrastructure Coordination Ecosystem: Services & Support (ACCESS) program, and National Artificial Intelligence Research Resource (NAIRR) Pilot for computational resources, including the Anvil supercomputer (National Science Foundation award OAC 2005632) at Purdue University and the Delta and DeltaAI advanced computing resources (National Science Foundation award OAC 2005572). The bird video frames used in Figure 1 and Figure 3 were adapted from the YouTube video "Videos for Cats to Watch" by Paul Dinning, and are used under fair use for academic and non-commercial purposes.

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

# Appendix

## A  Overview

In this supplemental document, we provide additional content that complements the main paper sections. Sec. B elaborates on additional details of the dataset curation. Sec. C provides additional experimental details, which include more model architecture details and evaluation information. Sec. D includes our ablation study. Sec. E provides additional visualization for teaser and qualitative model comparison results. Sec. F provides an extension to show the generated contents outside the first frame region in our model. Sec. G includes a discussion of the limitations of our model.

## B  Dataset Curation Details

The detailed filtering statistics can be found in Tab. 4. For faster curation, we switch the order between the camera filter and panoptic segmentation. This is because camera detection by CUT3R [64] spends much more time than image-based panoptic segmentation by Oneformer [21]. Though WebVid [4] shows a clearer and more direct concept for each video than other datasets, they have watermarks, and their resolution is low. Thus, we only consider around 1.5M videos of it to balance the dataset diversity and quality.

In the *Basic Filter*, we consider videos with at least 4 seconds, but not more than 20 seconds, with a frames per second (fps) range of $[20, 31]$. The aspect ratio of width to height must be larger than 1.35, which corresponds to 4:3 widely-used traditional metrics. The minimum pixel width is 400 pixels.

In the *Image Scoring*, we randomly select 2 images from each video and then get the score of these two images by executing multiple image-based assessment models. We sort the score from the smallest to the highest on each metric and filter based on the human subjective perspective for different datasets. Thus, the filtering strength is different between datasets based on their characteristics. Further, we consider the overlap scenarios between different scoring elements. We consider image quality assessment by ClipIQA [62], with the range lowest 3%, 5%, 15% filtered for OpenVid [38], VidGen [51], and WebVid [4] respectively. We consider text detection from an open-source repository EasyOCR [2], which is based on [2]. We sort the area size of the detected text and then filter the highest 15%, 10%, and 5%, respectively. We consider aesthetic assessment from NIMA [49] from the codebase of [8]. We filter the lowest 5%, 5%, and 10%, respectively. For the image complexity assessment, we use IC9600 [13] to evaluate the complexity score, where a lower value means less complexity, which lacks effective content to learn, and a higher value means more complexity, which has too many features to learn, like dense population or sightseeing. We filter both the lowest 10% and highest 5% for OpenVid, the lowest 5% and highest 10% for VidGen, and the lowest 5% and highest 10% for WebVid.

In the *Scene Cut*, we use TransNet V2 [46], which is based on the comparison results from Cosmos [1]. We filter any video that is detected with more than 1 scene. This ratio is not very high based on our observation. We think this is because the long video cases are already filtered initially.

In the *Camera Filter*, we use CUT3R [64], a state-of-the-art model for 3D reconstruction and camera pose estimation in dynamic video. However, evaluating 3D point positions jointly is computationally expensive, so we adopt the 224-resolution model and estimate camera poses over a 10-second window at 6 FPS. For each frame, we extract the predicted rotation, translation, and focal length. To sort videos based on camera motion intensity, we compute a score combining translational and rotational errors using the following formula:

$$\text{Score} = \|\mathbf{t}_1 - \mathbf{t}_2\|_2 + \cos^{-1}\left(\frac{\text{Tr}(R_1^\top R_2) - 1}{2}\right)$$

where $(\mathbf{t}_1, R_1)$ and $(\mathbf{t}_2, R_2)$ are camera poses between consecutive frames. Here, we filter the highest 40% of the rotation error, the highest 40% of the translation error, and the highest 10% of the focal length change. Additionally, we find that VidGen-1M exhibits significantly higher translational and rotational errors compared to WebVid and OpenVid, indicating more frequent and abrupt camera motion. Empirically, we observe that such unstable camera motion introduces

---

[2]https://github.com/JaidedAI/EasyOCR

Table 4: **Filtering process statistics across datasets.** Each row shows the number of videos retained and the percentage relative to the initial pool at that stage.

| Stage | OpenVid [38] | | VidGen [51] | | WebVid [4] | |
|---|---|---|---|---|---|---|
| | Count | Left Ratio (%) | Count | Left Ratio (%) | Count | Left Ratio (%) |
| Initial | 1.00M | 100.0% | 1.00M | 100.0% | 1.50M | 100.0% |
| Basic Filter | 537K | 53.7% | 821K | 82.1% | 1.276M | 85.1% |
| Image Scoring | 295K | 29.5% | 575K | 57.5% | 781K | 52.1% |
| Scene Cut | 280K | 28.0% | 518K | 51.8% | 483K | 32.2% |
| Panoptic Seg. | 155K | 15.5% | 396K | 39.6% | 218K | 14.5% |
| Camera Filter | 102K | 10.2% | 35K | 3.5% | 102K | 6.8% |
| Motion Filter | 86K | 8.6% | 32.5K | 3.3% | 82K | 5.5% |
| In-N-Out Filter | 29.7K | 3.0% | 9.2K | 0.9% | 33.4K | 2.2% |

ambiguity during training and degrades the success rate on the Frame In and the Frame Out intention we want. Therefore, we apply harsher filtering to the VidGen-1M subset.

In the *Panoptic Segmentation*, we consider 22 objects of COCO dataset [34] detected by One-Former [21] as the identity of interest, which includes *person, bicycle, car, motorcycle, airplane, bus, train, truck, boat, bird, cat, dog, horse, sheep, cow, elephant, bear, zebra, giraffe, sports ball, kite, and flower*. We want the identity to be more diverse than regular human face-oriented identity-preserving-to-video [90, 79, 33] (IP2V), but less than elements-to-video domain [12, 35] that consider all genre, either motionable or static objects. The maximum duration of the I-Frame adjustment mentioned in the main paper is only 5% of the full length. If the I-Frame index is farther away from this, we use the original frame index counted. We discard objects that are too small, less than 4% of the area, and too big, which occupy more than 40% area of the image. The number of points sampled from K-means ranges from $[12, 36]$ points based on the aforementioned area range. To avoid dense labeling with the same identity, we only allow at most 3 objects with the same label; otherwise, the video will be flittered.

In *Video Captioning*, there is no filtering. We apply QWen2.5-32B-VL-Instruct [73], which is the SOTA model for video captioning. We sampled only 1 frame per second with a resolution of $320 \times 448$ to save computation resources. The instruction text prompt we use is: *Please describe the video in 50 words. Only describe the temporal change of the video provided without describing the spatial information in the first frame provided. Only show the information with the highest confidence. Don't use any words like gesture, gesturing.* We apply captioning before Stage 1 motion-guided image-to-video generation training and the text prompt will be filtered with the rest of the procedure.

In *Motion Filter*, we first resize all videos to $384 \times 512$, which is the resolution used on the CoTracker demo [26]. We sampled our video to 24 FPS but stored the result in 12 FPS, which is our training dataset fetched FPS. The start frame is the frame index from the panoptic segmentation section and the end frame is 49 frames from it, which is also our training duration for each selected video clip. If the cycle tracking errors at the first frame are larger than 4% of the number of pixels in height, this tracking point will be filtered. If more than 33% of points of an object are filtered in this way, the entire object is not considered.

In *In-N-Out Filter*, we consider various aspect ratio of 16:9, 3:2, 4:3, 5:4, 1:1, and 4:5 with probability of 35%, 30%, 20%, 13%, 1%, 1% respectively, where the minimum height is 60%, 60%, 65%, 65%, 75%, and 85% of its original height respectively. The top-left position is randomly generated, and the code base will check if the selected box can fit in the image resolution. If not, this bounding box will be filtered, and consider the next one. We will do this way 2000 times. If there is no ideal bounding box found, this video is filtered. All video clip that starts from the index of panoptic segmentation will be considered. The SAM2 [42] will be utilized after at least one ideal bounding box is found. We use SAM2 to further refine the tracking points outlier cases. Since SAM2 is also expensive, we did not deploy SAM2 with the CoTracker3 [26] at the motion filter stage. Instead, we only use it at the final stage for final improvement. We further discard identity reference images that are less than 10% of the image resolution size based on SAM2 estimation.

Table 5: **Additional Conditioning comparisons to existing controllable video generation works.**

| Methods | Text | First Frame | Identity | Motion | Unbounded Canvas |
|---------|------|-------------|----------|--------|------------------|
| ConcatID [90] | ✓ | ✗ | ✓ | ✗ | ✗ |
| Direct-A-Video [74] | ✓ | ✗ | ✗ | ✓ | ✗ |
| ReVideo [37] | ✗ | ✗ | ✓ | ✓ | ✗ |
| VideoAnyDoor [53] | ✗ | ✗ | ✓ | ✓ | ✗ |
| Ours | ✓ | ✓ | ✓ | ✓ | ✓ |

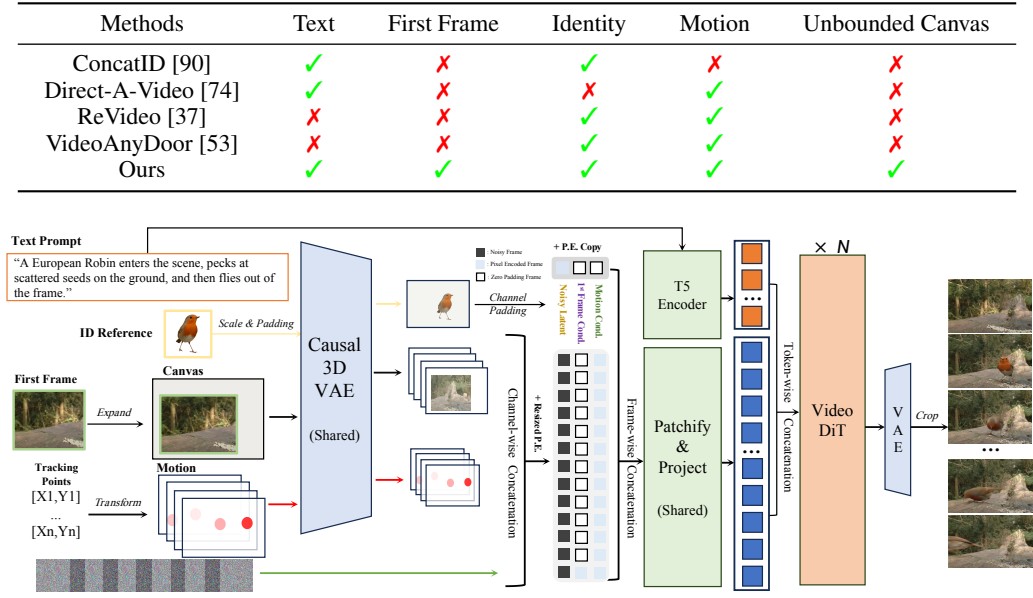

Figure 5: **Inference Pipeline.**

## C  Additional Experimental Details

In the motion control architecture, one-pixel coordinate as motion images form is insensitive for the VAE encoder to effectively encode and decode; thus, similar to previous works [67, 58], we enlarge pixels to a square rectangular box and then execute a 2D dilation algorithm from deblurring domain to increase the perceptibility. The rectangular box border length is 6 pixels for a video with an original height of 384, which means that the size will be scaled with the exact height. Further, to accommodate sparse motion points (1-2 points per object) provided by the user side during the inference, we randomly drop points during the training. The number of tracking points is randomly dropped with the probability of 33% and 60% for each stage respectively. Different from optical flow-based motion control works [67, 58, 87], we do not need a padding frame on the last frame to align the total number of frames, where all of our motion conditioning frames represent a spatiotemporal pixel-aligned motion status.

Our inference pipeline can be found in Fig. 5. The model will start from pure noise with the size of the canvas region. The first frame will be expanded to the canvas size by the $\mathbf{B}_{canvas}$ setting for the top-left and bottom-right expansion pixel number. The motion is conditioned on the canvas size as a whole after converting from the raw pixel spatiotemporal coordinates. Similarly, the ID reference images will be scaled and padded to the same size as the canvas size. Similar to training, after forwarding the video DiT in each timestep, the text tokens and ID tokens will be discarded, but the video DiT in the next timestep will be conditioned on fresh text and ID tokens again. The denoised latent will be decoded by the pre-trained VAE decoder, and then we cropped the pixel space videos to be the exact same shape as the first frame based on $\mathbf{B}_{canvas}$ setting.

In stage 2 training, we randomly drop ID reference images with a probability of 15% in the training, where we only have Frame Out scenarios to consider. The ID reference position will be a placeholder with a monocular white color. This is also how we get our stage 2 Frame Out results in Tab.2 of the main paper. After token-wise concatenation, the order of tokens should be text tokens, video tokens, and then the ID reference tokens accordingly.

Since the resolution and the number of frames that each baseline model can generate are different in the quantitative comparisons. For a fair comparison, we first resize the validation dataset to the supported default resolution of each baseline method. Then, we resize the generated videos to the uniform resolution of 256×384 with the number of frames of 14 and 49 respectively for the Frame Out and Frame In table. This number of frames represents the minimum frame number across all

models in that table. For the SkyReels-A2 [12], based on their official code implementation, we generate 81 frames and then crop the first 12 frames since the first 12 frames look highly distorted. We only compare the remaining 69 frames.

Works like ReVideo [37] and VideoAnyDoor [53] and other similar works [92, 6] are conditioned on the full video sequence as inputs, not just the first frame, which means that they are video-to-video generation instead of image-to-video generation we target at. Furthermore, these works are more focused on the insertion and editing in the middle instead of entering and leaving effects we pursue. Thus, we believe that this direction is not an ideal choice for Frame In and Frame Out baselines. An additional condition comparison table can be found in Tab. 5.

Base generative metrics evaluation description:

**FID** [18] measures the distance between the distribution of generated and real frame features, extracted using an Inception network. It is sensitive to both image quality and diversity, and lower scores indicate better generation.

**FVD** [55] extends FID to the temporal domain, computing the Fréchet distance between distributions of video-level features extracted using a pre-trained I3D model. It captures spatiotemporal coherence in generated videos and is similarly lower-is-better.

**LPIPS** [83] quantifies the perceptual similarity between individual frames and their references using deep feature distances from a pre-trained network. Unlike pixel-wise metrics, LPIPS correlates well with human judgment of visual similarity.

The prompt we used for motion description for SkyReels-A2 [12] and Phantom [35] is: *Please describe the video in 50 words. Describe the motion of the object clearly and in details, but in the natural and direct language. It is expected that an object will enter and/or exit the image. Describe how the character is moved in and exit, like from the top, left, right, bottom.*

Table 6: **Human Study with Baseline Methods on Win Rate**.

| Method | Win Rate↑ |
|---|---|
| DragAnything [69] | 83.3% |
| ImageConductor [32] | 100.0% |
| ToRA [87] | 73.3% |
| Phantom [35] | 91.1% |
| SkyReels-A2 [12] | 92.2% |

**Human Study**. Further, we conducted a small-scale user study as shown in Tab. 6. We randomly selected 30 videos from our evaluation benchmark set and asked three anonymous human raters to perform pairwise comparisons between our model and baseline methods. The instruction we gave is: *You are be shown two generated videos from different models. You need to choose one that appears clearer and better Frame Out and Frame In effect. Meanwhile, we also provide the GroundTruth Motion and BoundingBox information. For the FrameIn cases, you will be given with extra ID reference condition. Frame Out means that an object exists in the first frame condition and is leaving the scene. Frame In means that an identity reference that does not first exist in the first frame, appears in the scene naturally.*

In each comparison, the preferred video that achieves a better Frame In and Frame Out effect received a score of 1; the other, 0. A total of 450 comparisons were collected. We report the win ratio, defined as the number of wins divided by the number of comparisons, as a measure of user preference. Higher values indicate a stronger preference for our model. The results align well with the automatic metrics reported in the paper, such as trajectory error and VLM evaluation. The consistency across multiple metrics also provides supporting evidence for the statistical significance of our gains over baselines.

## D   Ablation Study

As shown in Tab. 7, we compare several different purpose-trained models under different settings. Due to the computation limitations, we train all models with 12K iterations of a batch size of 8. The training is done on stage 2, and all utilize the same pre-trained stage 1 weight (including the baseline).

Table 7: **Ablation Study on Frame In Comparisons**.

| Method | FID↓ | FVD↓ | LPIPS↓ | Traj. Err.↓ | VSeg MAE↓ | Rel. DINO↓ | VLM↑ |
|---|---|---|---|---|---|---|---|
| Baseline | 30.18 | 283.43 | 0.219 | 9.54 | 0.0107 | 0.68 | 0.868 |
| 384x480 Inference | 29.73 | 249.34 | 0.223 | 9.48 | 0.0120 | 1.10 | 0.797 |
| No Full Filed Loss | 30.74 | 278.12 | 0.238 | 48.49 | 0.0497 | 1.92 | 0.792 |
| New Absolute PE | 36.03 | 299.17 | 0.247 | 10.21 | 0.0114 | 0.97 | 0.836 |

We compare all models on the Frame In evaluation, which is the most representative task for all conditions to be considered.

**Inference Resolution Influence**. In the training, we train a fixed canvas size of 384x480, but we test at 448x640 as our baseline. We also test at 384x480 to see if the resolution in the inference influences the conclusion. We can see from the second row of Tab. 7 that some metrics are higher and some are lower. Overall, the inference resolution does not provide a direct advantage to the final numerical results.

**Full Field Loss Influence**. We also provide a model that is not trained with full field loss. This means that the ground-truth target latent provided only has the encoded information from the first frame region instead of the canvas region in the baseline. The region outside the first frame is padded with zero. It is worth noting that the comparison of FID, FVD, LPIPS, Relative DINO, and VLM is only for the first frame region, which means that the padded zero is not included in the evaluation. As we can see from the third row of the Tab. 7, almost all metrics dropped. Here, trajectory error, video segmentation MAE, and relative DINO dropped evidently. We believe that the introduction of full field loss is significant to the final visual generated quality and motion consistency.

**Absolute Position Encoding Influence**. In our baseline model, we train the model with resized absolute position encoding by trilinear interpolation. Despite this method, another solution to embrace different resolution inputs for both training and inference could be to create a new position encoding each time, where the position encoding for the identity reference is also refreshed each time. However, as shown in the fourth row of Tab. 7, this will lead to a performance drop in all metrics, especially FID and relative DINO. We believe that reusing the learned position encoding by applying a simple trilinear interpolation is helpful for the versatile video resolution inputs, which maintain similar data distribution from the learned fixed position embeddings at the minimum cost.

# E   Additional Visualization

A more complete sequence of the generated videos and all conditions of the teaser can be found in Fig. 6. Further, we provide more of our generated video samples in Fig. 7. Some demo images or ID references are chosen or cropped from Davis Dataset [41] and online images. We show multiple different kinds of combinations of Frame In and Frame Out. For the *physical interaction*, we mean the interaction caused by breaking new ID references to the object that already exists in the first frame. We set all the height and width to be a multiplier of 32 due to the VAE limitation. As shown in Fig. 6 and Fig. 7, the setting for the height and width of the canvas and first frame region can be versatile in aspect ratio and size while generating high fidelity and stable results. Further, we empirically found the generation is stable as long as the canvas height is less than 480 pixels and the canvas width is less than 720 pixels, which is the training resolution for our pre-trained base model, CogVideoX [76].

As shown in Fig. 8 and Fig. 9, we also present extra qualitative comparisons with the baseline methods. For the Frame In comparison, we can see that baselines like Phantom [35] and SkyReels-A2 [12] cannot understand the Frame In intention we want. The identity reference either already existed in the first frame (cases 2 and 3) or never came to the scene (case 1). Further, the first frame condition is not faithfully considered as the main condition. The scene they generate is similar in elements but different in objects, 3D position, and physical relationship. On the contrary, ours shows clear alignment with the motion conditioning and reliable faithfulness to the first frame. For the Frame Out comparison, we can see that DragAnything [69] might have exaggerated motion when the motion conditioning outside the first frame region is not provided (case 1). Image Conductor [32] cannot faithfully generate the videos. ToRA [87] does not provide the stable Frame Out effect we want.

## F    Full Canvas Generation Extension

The full field loss implementation makes the generative objective closer to the outpainting model design; however, compared to video outpainting works like [60, 10], which needs full sequence video as inputs and converges to a video-to-video generation task, ours only provides the first frame and focuses on the balance of motion identity-preserving conditioning. Thus, we believe that it is appropriate to name our conditioning as *unbounded canvas*, instead of the vanilla outpainting. In Fig. 10, we provide the visual content outside the first frame region for two teaser image examples. This is the byproduct of our full field loss implementation, which jointly generates the full canvas region compared with the ground-truth latent in the training. We observe that some examples will show unwanted color distortion for the region outside the first frame region. Since our area of interest is always the first frame region, we cropped with only the first frame region in the inference. We leave this problem to future works.

## G    Limitation

Our method shows promising results, but there are nevertheless some limitations that are worth sharing, as shown in Fig. 11. Mainly, this lies in the 3D ambiguity when there is only one point for the motion trajectory. We have to restate that most current works on motion control are single trajectory point-based. One trajectory point for breaking new ID reference information is hard to present the pose ambiguity (see Fig. 11 (c)). Sometimes we want to generate the back view, but we may see the side view cases. Further, one point trajectory is hard to control the size of the ID reference, where sometimes it might be bigger (see Fig. 11 (d) or smaller (see Fig. 11 (e)) than expected. Further, the camera motion lies in the pre-trained model dataset [76], and our filtering method from CUT3R [64] cannot completely remove all videos with camera motion. This leads our model to generate videos with some unwanted camera motion (see Fig. 11 (b)). These issues might be solved by introducing a more robust 3D control system, like camera control or size control for the ID reference.

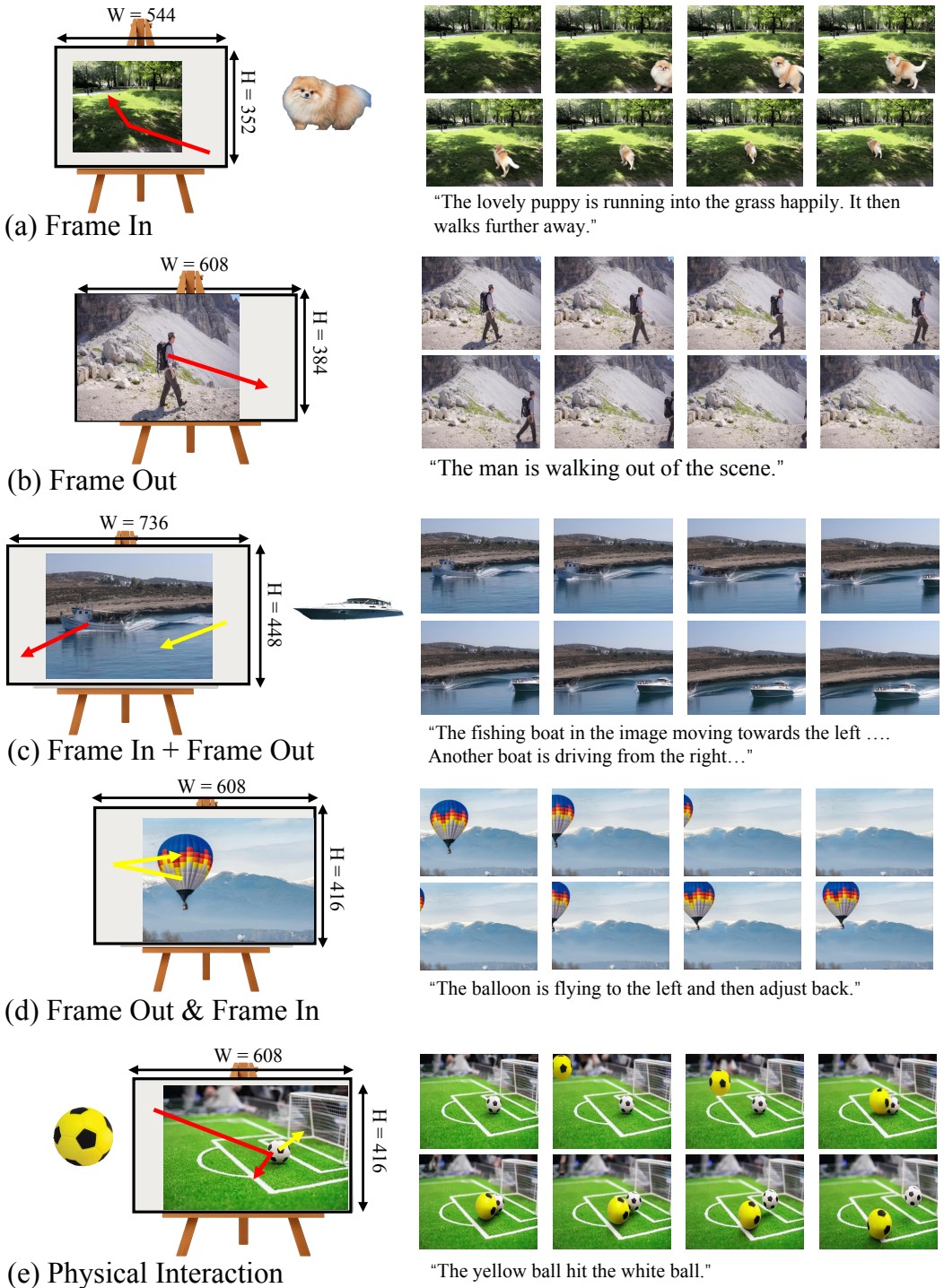

(a) Frame In

"The lovely puppy is running into the grass happily. It then walks further away."

(b) Frame Out

"The man is walking out of the scene."

(c) Frame In + Frame Out

"The fishing boat in the image moving towards the left …. Another boat is driving from the right…"

(d) Frame Out & Frame In

"The balloon is flying to the left and then adjust back."

(e) Physical Interaction

"The yellow ball hit the white ball."

Figure 6: **Detailed Conditioning for Generated Videos on Teaser Image.**

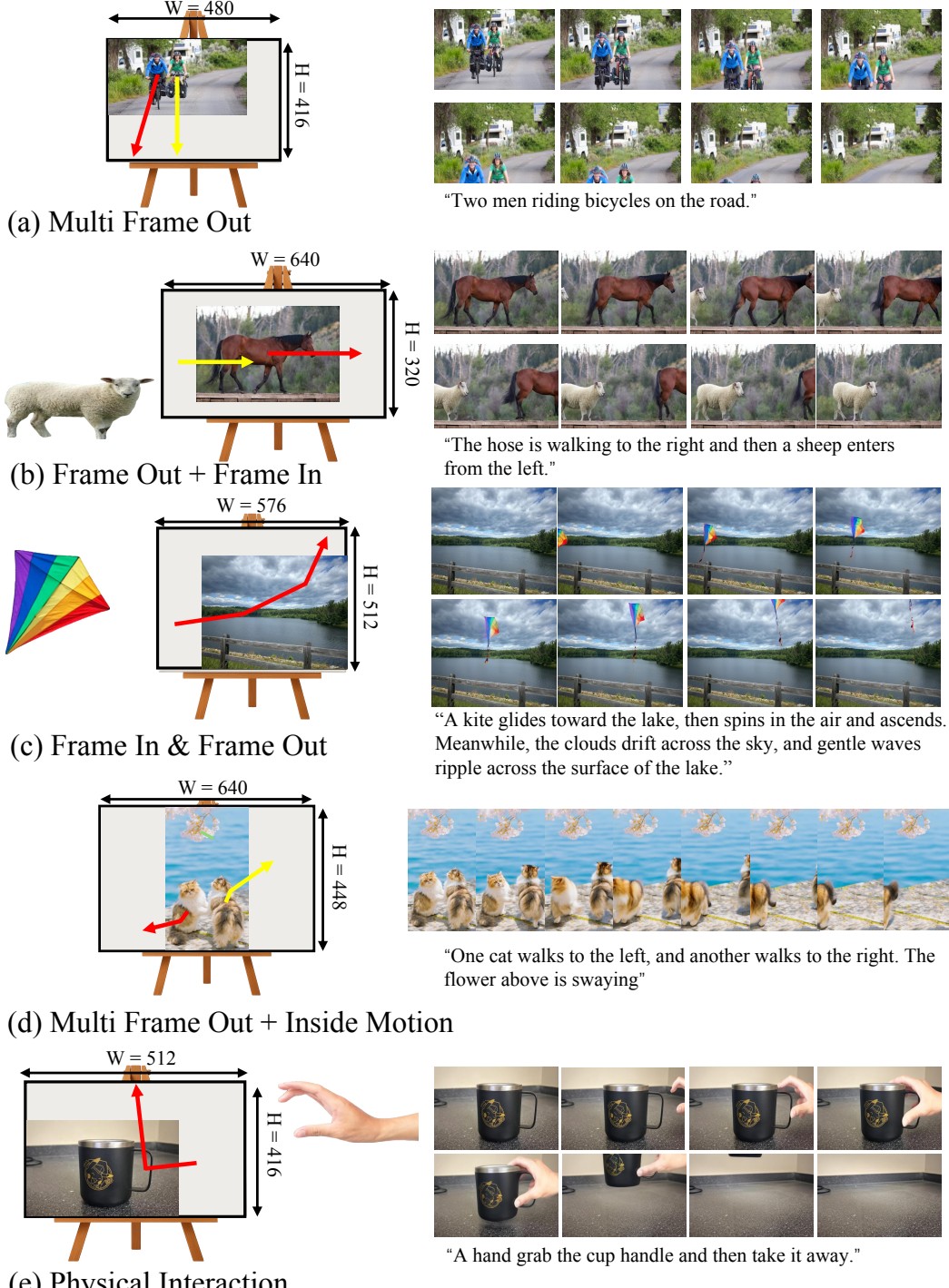

(a) Multi Frame Out

"Two men riding bicycles on the road."

(b) Frame Out + Frame In

"The hose is walking to the right and then a sheep enters from the left."

(c) Frame In & Frame Out

"A kite glides toward the lake, then spins in the air and ascends. Meanwhile, the clouds drift across the sky, and gentle waves ripple across the surface of the lake."

(d) Multi Frame Out + Inside Motion

"One cat walks to the left, and another walks to the right. The flower above is swaying"

(e) Physical Interaction

"A hand grab the cup handle and then take it away."

Figure 7: **More Generated Examples for Our Method.**

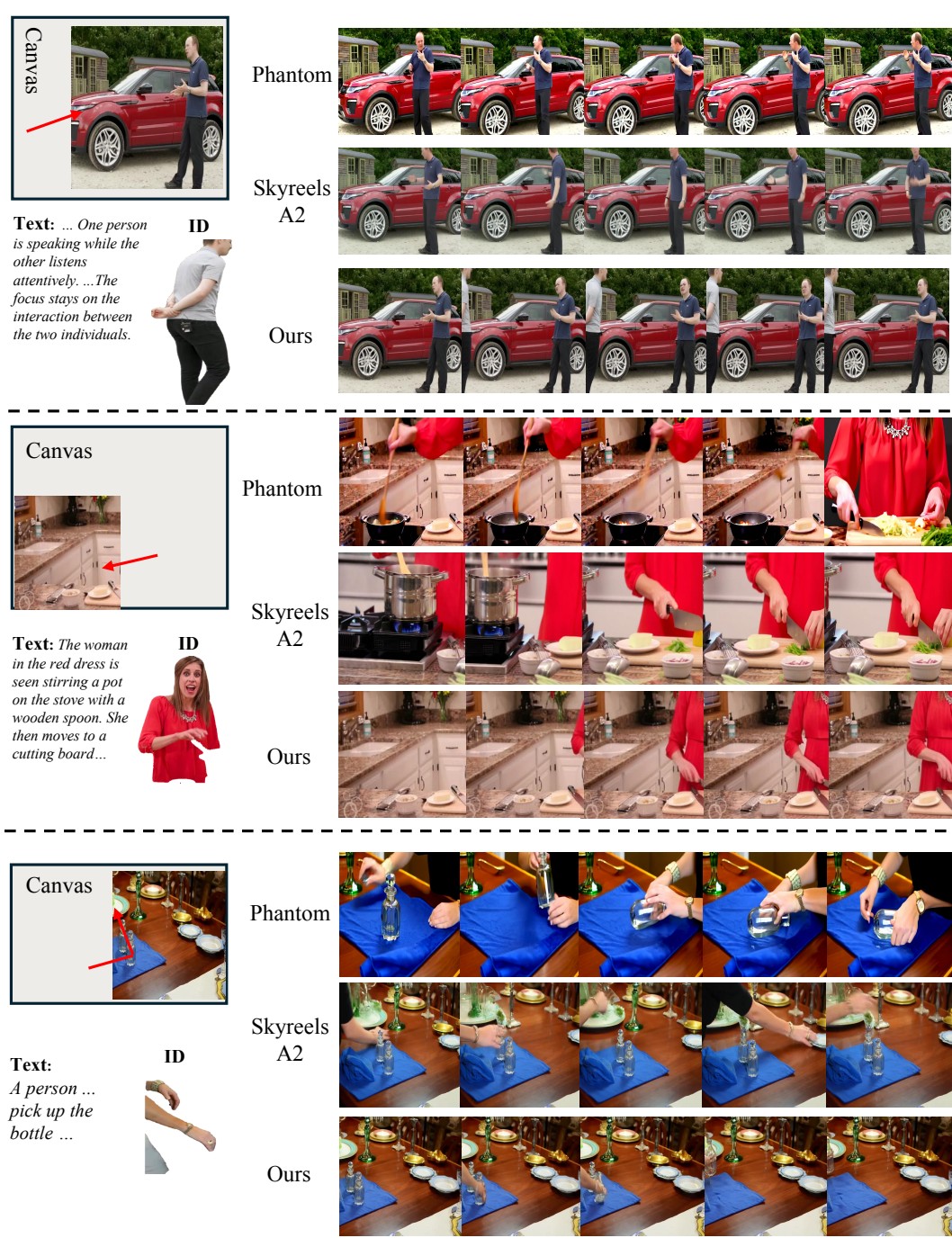

Figure 8: **Extra Frame In Comparisons with Baseline Methods**.

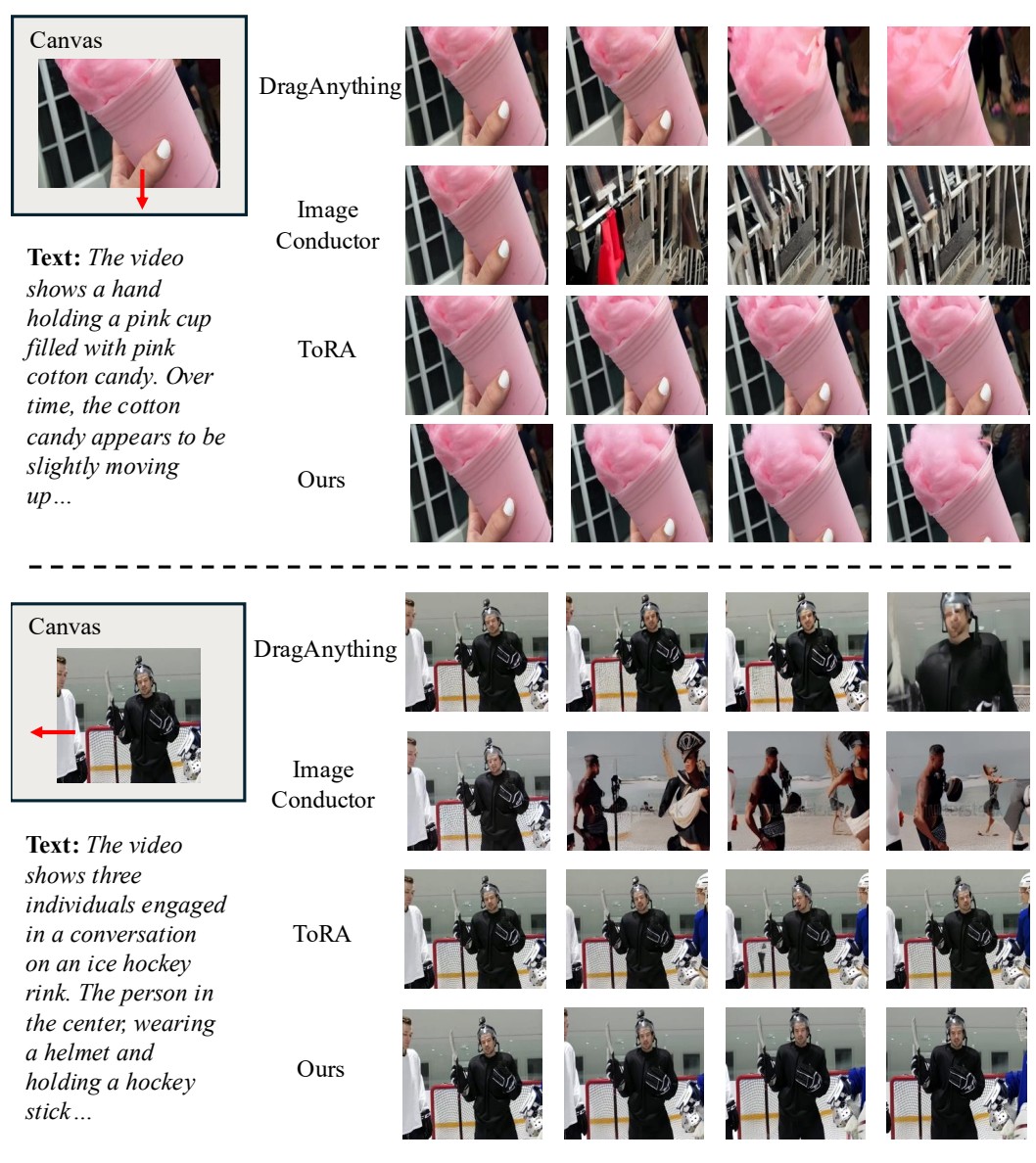

Figure 9: **Extra Frame Out Comparisons with Baseline Methods**.

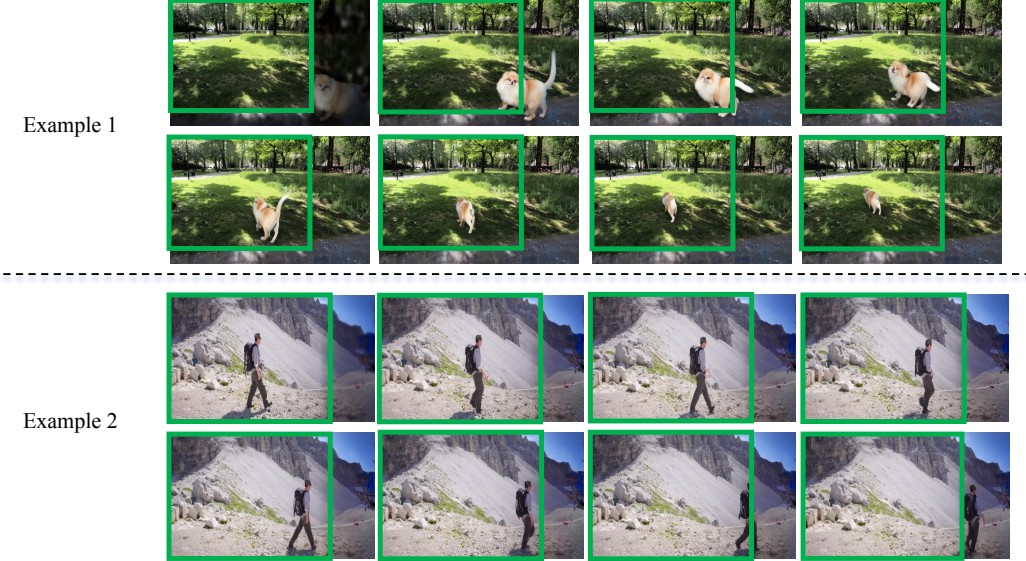

Figure 10: **Full Canvas Generation Showcase.** We pick two examples from the teaser image to show the generated results of our unbounded canvas generation outside the first frame region. The green bounding box is the eventual generated video that will be left in the inference.

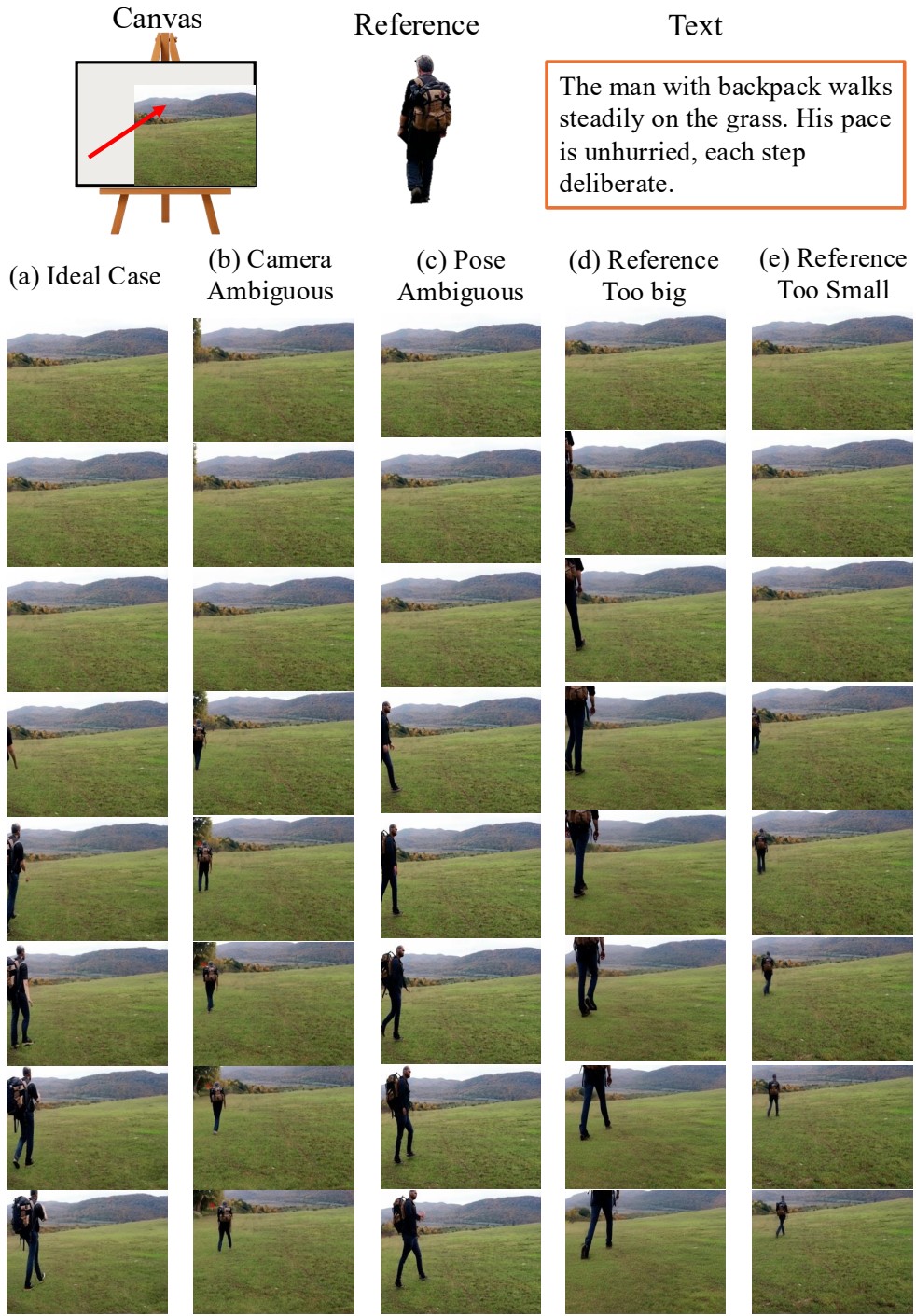

Figure 11: **Limitations.** Given the same input and conditions, our model may produce different outputs under different random seeds. We consider (a) to be an ideal case and illustrate several limitations: camera motion ambiguity in (b), ID reference pose ambiguity in (c), overly large reference objects in (d), and overly small reference objects in (e).

