# OpenReview forum: "Frame In-N-Out: Unbounded Controllable Image-to-Video Generation"
_NeurIPS.cc/2025/Conference — NeurIPS 2025 poster_

### Official Review · Reviewer_tVWT · 2025-06-26

**Clarity:** 3
**Significance:** 3
**Originality:** 2
**Rating:** 4
**Confidence:** 4

**Summary:**

This paper introduces Frame In-N-Out, a novel task in video generation. It adapts a video diffusion transformer to incorporate conditional inputs and curate a dedicated dataset along with evaluation benchmarks. Experimental results demonstrate that the proposed method outperforms existing baselines.

**Questions:**

Why is random noise added to the ID reference during training? Given that inference only uses either zero noise or full noise, wouldn’t it be more consistent to align the training scheme with these two discrete noise levels?

**Ethical Concerns:**

["NO or VERY MINOR ethics concerns only"]

**Final Justification:**

Most of my concerns have been addressed. However, I still believe the primary contribution of this work comes from the proposed dataset. And the contribution in terms of methods is limited. Therefore, I am maintaining my original score of borderline accept.

**Limitations:**

yes

**Paper Formatting Concerns:**

I don't have any concern about this.

**Quality:**

3

**Strengths And Weaknesses:**

Strengths:
1. The paper is well written and easy to follow.
2. The proposed dataset and evaluation benchmark can be valuable for the community.

Drawbacks:
1.  The novelty is somewhat limited in the architecture, as the method primarily adapts an existing Diffusion Transformer to accommodate conditional inputs. Given this, the core contribution appears to be the proposed dataset and evaluation benchmark—making the work potentially better suited for a dataset/benchmark track rather than the main conference.
2. There is a resolution mismatch between training (384 × 480) and inference (480 × 720), which may negatively impact model performance. To address this, I recommend adding a final training stage using either high-resolution or mixed-resolution data.
3. The curated dataset lacks videos with significant camera motion, limiting the model's ability to handle Frame In/Out and camera control simultaneously. Can the current pipeline be extended to support camera motion?

---

> ### Author Rebuttal · Authors · 2025-07-31
>
> Dear reviewer tVWT,
>
> Thank you for your careful reading of the manuscript and constructive review of our work. Below, we address your questions and the weaknesses mentioned:
>
> - **[W1]: The novelty is somewhat limited in the architecture, as the method primarily adapts an existing Diffusion Transformer to accommodate conditional inputs. Given this, the core contribution appears to be the proposed dataset and evaluation benchmark—making the work potentially better suited for a dataset/benchmark track rather than the main conference.**
>
>     Thank you for the thoughtful feedback. While our model builds upon the DiT backbone, the core novelty lies in the unified integration of multiple conditional inputs—motion trajectories, identity references, and unbounded spatial context—through a coherent architectural design and a novel data curation pipeline. To our knowledge, this is the first work to jointly enable all three aspects under the Frame In and Frame Out setting. Architecturally, we introduce channel-wise concatenation to reduce computation for pixel-aligned conditioning, and token-wise fusion to stabilize training with multiple inputs. Our focus is not only on the dataset, but also on demonstrating a controllable video generation framework that balances implementation simplicity, computational efficiency, and novel generative capabilities. For these reasons, we believe the main conference track remains the most appropriate venue for this work.
>
> - **[W2]: There is a resolution mismatch between training (384 × 480) and inference (480 × 720), which may negatively impact model performance. To address this, I recommend adding a final training stage using either high-resolution or mixed-resolution data.**
>
>     Thank you for the insightful suggestion. We agree that the resolution gap between training and inference could impact performance. As part of our ongoing work, we are incorporating a final training stage with higher-resolution videos, as you suggested. In addition, we are experimenting with larger base models beyond CogVideoX-5B to further enhance quality. We plan to include these updated models in the camera-ready version once training is complete.
>
> - **[W2]: The curated dataset lacks videos with significant camera motion, limiting the model's ability to handle Frame In/Out and camera control simultaneously. Can the current pipeline be extended to support camera motion?**
>
>     Thank you for the great question. Camera motion control is indeed feasible within our framework and was part of our original vision. However, we found that obtaining high-quality videos with accurate 3D camera motion remains challenging. Even with advanced models like CUT3R, estimates of rotation, translation, and camera intrinsics are still unreliable on in-the-wild videos. Given these limitations, we chose to focus first on Frame In-N-Out motion control as a foundational step, and we aim to incorporate camera motion in future work.
>
> ---
> Questions:
> ---
>
> - **[Q1]: Why is random noise added to the ID reference during training? Given that inference only uses either zero noise or full noise, wouldn’t it be more consistent to align the training scheme with these two discrete noise levels?**
>
>     Thank you for pointing this out. We initially adopted this training setup following Concat-ID [80]. However, our recent experiments indicate that this mismatch does not lead to significant changes in performance. We will revise the manuscript to clarify this and better align the explanation with our findings.

---

> > ### Comment · Reviewer_tVWT · 2025-08-05
> >
> > Thank you for your response. Most of my concerns have been addressed. However, I still believe the primary contribution of this work comes from the proposed dataset. And the contribution in terms of methods is limited. Therefore, I am maintaining my original score of borderline accept.

---

### Official Review · Reviewer_4Tsm · 2025-07-01

**Clarity:** 4
**Significance:** 2
**Originality:** 3
**Rating:** 5
**Confidence:** 5

**Summary:**

This paper proposed a novel video generation problem called "Frame In" and "Frame out", which an desired object can enter the scene or leave the scene, with fully controllable trajectory. A dataset and a evaluation protocol is designed for this task. The authors also proposed an initial attempt towards this problem. Leveraging an efficient identity-preserving motion-controllable video Diffusion Transformer architecture, The proposed method outperform baselines in the designed evaluation protocol.

**Questions:**

Line 287 - 289 "Vision Language Model evaluation (VLM) utilizes a modern large vision language model, Qwen 2.5 VL 32B [65], with full video sequences and an instruction prompt to justify if the object gets out of the first frame or enter the first frame. We evaluate the ratio of correctness compared to the returns with GT video inputs. This metric is intended to align the overall subjective success rate analysis."

VLMs can be unreliable and sensitive to prompt phrasing, raising concerns about consistency and reproducibility. Why not use more interpretable alternatives like object tracking or mask-based analysis? Would a hybrid evaluation be more robust?

**Ethical Concerns:**

["NO or VERY MINOR ethics concerns only"]

**Final Justification:**

The paper focused on a narrow problem in controllable video generation, and it could be potentially useful in CGI applications. The design and experiments are sound, and the rebuttal also addressed my concern. Therefore I have my score adjusted.

**Limitations:**

The authors have addressed technical limitations reasonably well, but a more thorough discussion of potential societal risks and ethical considerations would strengthen the paper. Particularly given the realism and identity control aspects of the generated videos. E.g. Deepfake-like misuse (inserting people into scenes realistically)?

**Paper Formatting Concerns:**

/

**Quality:**

3

**Strengths And Weaknesses:**

S1) This paper is well-written and easy to follow, with a clear motivation and problem formulation. It serves as a strong example of work that introduces a novel and meaningful problem setting.

S2)  The technical design is sound and informative, with clear architectural choices and efficient integration of motion, identity, and unbounded spatial control. Although no empirical ablation is provided, the proposed framework appears adaptable to a wide range of current I2V architectures, which speaks to its generality.

W1) While the Frame In-N-Out scenario is novel, it’s a narrow subset of controllable video generation, building upon prior tasks like identity -referenced video generation and trajectory / motion-controlled video generation.

W2) The designed automated metrics looks valid to me. But given the subjective nature of video realism and controllability, a small-scale user study would strengthen the evaluation.

---

> ### Author Rebuttal · Authors · 2025-07-31
>
> Dear review 4Tsm,
>
> Thank you for your careful reading of the manuscript and constructive review of our work. Below, we address your questions and weaknesses mentioned:
>
> - **[W1]: While the Frame In-N-Out scenario is novel, it’s a narrow subset of controllable video generation, building upon prior tasks like identity-referenced video generation and trajectory / motion-controlled video generation.**
>
>     We agree that our work builds on prior tasks. The novelty lies in unifying identity-preserving and motion-controlled generation within a single framework to achieve the underexplored cinematic concept of Frame In and Frame Out. While trajectory-conditioned models typically rely on a first-frame input and identity-conditioned models often generate content from null space, our approach integrates both through an unbounded canvas design. This enables conditioning simultaneously on first-frame appearance, motion trajectories, and extended spatial context. To our knowledge, this unified formulation has not been explored in prior works. We will clarify this in the final version.
>
>
> - **[W2]: Given the subjective nature of video realism and controllability, a small-scale user study would strengthen the evaluation.**
>
>     Thank you for the thoughtful suggestion! To address this concern, we conducted a small-scale user study as part of this rebuttal. We randomly selected 30 videos from our evaluation benchmark set and asked three anonymous human raters to perform pairwise comparisons between our model and baseline methods. In each comparison, the preferred video received a score of 1; the other, 0. A total of 450 comparisons were collected. We report the win ratio, defined as the number of wins divided by the number of comparisons, as a measure of user preference. Higher values indicate a stronger preference for our model. The results align well with the automatic metrics reported in the paper, such as trajectory error and VLM evaluation. The consistency across multiple metrics also provides supporting evidence for the statistical significance of our gains over baselines. We will include this human study and its details in the final version.
>
>     | Method         | Win Rate (Higher better) |
>     |----------------|--------------------------|
>     | DragAnything   | 83.3%                    |
>     | ImageConductor | 100%                     |
>     | ToRA           | 73.3%                    |
>     | Phantom        | 91.1%                    |
>     | SkyReels-A2    | 92.2%                    |
>
> ---
> Questions
> ---
>
> - **[Q1]: VLMs can be unreliable and sensitive to prompt phrasing, raising concerns about consistency and reproducibility. Why not use more interpretable alternatives like object tracking or mask-based analysis? Would a hybrid evaluation be more robust?**
>
>     We completely agree, and in fact, we have provided more interpretable alternatives in the manuscript, such as object tracking evaluation (Lines 274-280) and mask-based analysis (Lines 281-285). These evaluations provide complementary perspectives, and the results are consistent across all metrics.
>
>
> - **[Q2]: A more thorough discussion of potential societal risks and ethical considerations would strengthen the paper.**
>
>     Thank you for highlighting this important point. We fully agree and will include the following discussion in our final version:  While our technology offers promising applications in filmmaking, education, and creative fields, it may also pose risks if misused, for example, in generating inappropriate or unauthorized content. We emphasize the importance of obtaining proper consent when using personal images or identity references, and we encourage users to respect copyright, privacy, and cultural norms relevant to their regions.

---

> ### Comment · Reviewer_4Tsm · 2025-08-04
>
> I find the table in your W2 rebuttal is a bit confusing. "Method" should be "Ours v.s. Method"? Please also report the inter-rater agreement (e.g. fleiss kappa), since you have three anonymous human raters on each sample.
>
> Overall the rebuttal is convincing enough. I have adjusted the score.

---

> > ### Author Response · Authors · 2025-08-07
> >
> > Thank you for the suggestion. You’re right! The table header should read “Ours vs. Method”;
> > We report the agreement ratio among the three raters, which is defined as the proportion of rater-rater pairs that selected the same option out of all possible rater-rater pairs.
> >
> > For Ours vs. ToRA, the ratio is 71%.
> >
> > For Ours vs. DragAnything, the ratio is 82%.
> >
> > For Ours vs. ImageConductor, the ratio is 100%.
> >
> > For Ours vs. Phantom, the ratio is 95.5%.
> >
> > For Ours vs. SkyReels, the ratio is 95.5%.
> >
> > We will update the table and report the inter-rater agreement in the final version.

---

### Official Review · Reviewer_XTCB · 2025-07-02

**Clarity:** 3
**Significance:** 2
**Originality:** 3
**Rating:** 4
**Confidence:** 4

**Summary:**

Frame In-N-Out introduces a novel paradigm for controllable image-to-video generation that enables objects to move beyond the spatial boundaries of the initial frame. The approach supports two key cinematic techniques:
- Frame Out: Existing objects in the first frame move completely outside the visible bounds
- Frame In: New identity objects enter the scene from outside the frame boundaries
The method extends the conditioning canvas beyond the first frame region, uses a two-stage training approach, and introduces specialized evaluation metrics for this new task. The authors build upon CogVideoX-I2V architecture with modifications for unbounded motion control and identity preservation.

**Questions:**

See Strength & Weakness

**Ethical Concerns:**

["NO or VERY MINOR ethics concerns only"]

**Final Justification:**

I will keep my positive rating of the paper, but it's borderline-level. I'm also ok if it's not accepted.

**Limitations:**

See Strength & Weakness

**Paper Formatting Concerns:**

No Paper Formatting Concerns.

**Quality:**

3

**Strengths And Weaknesses:**

Strength
- Novel Problem Formulation
 This work is the first to explicitly tackle “Frame In–Out” patterns in video generation, addressing a gap in the literature and offering a new paradigm for cinematic editing tasks.
- Comprehensive System Design
 The authors present an end-to-end pipeline—from semi-automatic data curation through two-stage model training to thorough evaluation—ensuring that each component is well integrated and motivated by practical production needs.
- Rigorous Data Curation
 Dataset creation combines automated methods with human quality control; a cycle-tracking pipeline verifies annotation accuracy, and intelligent pattern recognition reliably isolates Frame In and Frame Out scenarios.
- Thoughtful Architectural Choices
 Key design elements (an unbounded canvas, full-field loss, and efficient channel concatenation) are carefully chosen to support flexible spatial generation and maintain consistency across frames.
- Strong Experimental Validation
 The method is benchmarked against both motion-controllable and elements-to-video baselines using diverse metrics—including novel task-specific measures like VSeg MAE and Trajectory Error—demonstrating clear quantitative gains.


Weakness
- Limited Technical Novelty
 Many architectural ideas (e.g., canvas expansion) are incremental adaptations of existing Diffusion Transformer (DiT) frameworks and resemble prior outpainting or “Follow-Your-Canvas” methods.
- Evaluation Constraints
 The test set is small (183 Frame In and 189 Frame Out examples), lacks human subjective studies for cinematic quality, and omits statistical significance testing to bolster confidence in reported gains.
- Scalability and Practical Concerns
 A fixed low training resolution (384×480) limits real-world applicability at higher resolutions; the two-stage training pipeline adds complexity and computational overhead, and memory requirements for varying canvas sizes are unspecified.
- Methodological Gaps
 The canvas expansion strategy is under-motivated from a theoretical standpoint, identity preservation relies on older techniques rather than recent advances (e.g., ConsisID), and there are few ablation studies on critical design choices like full-field loss or training stages.

---

> ### Author Rebuttal · Authors · 2025-07-31
>
> Dear reviewer XTCB,
>
> Thank you for your careful reading of the manuscript and constructive review of our work. Below, we address your questions and weaknesses mentioned:
>
> - **[W1]: Many architectural ideas (e.g., canvas expansion) are incremental adaptations of existing Diffusion Transformer (DiT) frameworks and resemble prior outpainting or “Follow-Your-Canvas” methods.**
>
>     We agree that our model builds on existing components such as DiT. Our contribution lies in the integration of these elements and a novel data curation pipeline to realize the underexplored cinematic concept of Frame In and Frame Out. Unlike video outpainting methods like Follow-Your-Canvas or Be-Your-Outpainter, which operate on full video sequences in a video-to-video manner, our framework requires only the first frame as input and emphasizes motion control and identity preservation. Additionally, while outpainting focuses on generating content beyond the original canvas, our goal is to model and animate content within the original image region. We will further clarify these distinctions in the final version.
>
>
> - **[W2]: The test set is small (183 Frame In and 189 Frame Out examples), lacks human subjective studies for cinematic quality, and omits statistical significance testing to bolster confidence in reported gains.**
>
>     Thank you for pointing this out. To address this concern, we conducted a small-scale user study as part of this rebuttal. We randomly selected 30 videos from our evaluation benchmark set and asked three anonymous human raters to perform pairwise comparisons between our model and baseline methods. In each comparison, the preferred video received a score of 1; the other, 0. A total of 450 comparisons were collected. We report the win ratio, defined as the number of wins divided by the number of comparisons, as a measure of user preference. Higher values indicate a stronger preference for our model. The results align well with the automatic metrics reported in the paper, such as trajectory error and VLM evaluation. The consistency across multiple metrics also provides supporting evidence for the statistical significance of our gains over baselines. We will include this human study and its details in the final version.
>
>     | Method         | Win Rate (Higher better) |
>     |----------------|--------------------------|
>     | DragAnything   | 83.3%                    |
>     | ImageConductor | 100%                     |
>     | ToRA           | 73.3%                    |
>     | Phantom        | 91.1%                    |
>     | SkyReels-A2    | 92.2%                    |
>
>
>
> - **[W3-1]: A fixed low training resolution (384×480) limits real-world applicability at higher resolutions.**
>
>     We agree that the low training resolution is a limitation, largely due to the computational constraints common in academic settings. That said, our model supports arbitrary output resolutions at inference time, enabled by the interpolation of positional embeddings (Lines 165–171). As illustrated in Supplementary Figures 2 and 3, the generated video resolutions vary across examples. We plan to scale our training to higher resolutions in future work.
>
>
> - **[W3-2]: The two-stage training pipeline adds complexity and computational overhead, and memory requirements for varying canvas sizes are unspecified.**
>
>     We fully agree with the reviewer that one-stage training offers practical benefits. That said, multi-stage training remains a widely adopted strategy to improve training stability and serve as stage-wise checks, especially in recent controllable video generation works such as Image Conductor [28], Phantom [31], MotionCtrl [59], and ToRA [77]. This is particularly useful for models with complicated multi-condition inputs, as in our case. For reference, generating a 49-frame video at 448×640 resolution takes approximately 16 GB of memory at inference time. The training memory cost is 57GB with 8bit Adam. We will report more details about computational overhead in the final version.
>
>
>
> - **[W4-1]: The canvas expansion strategy is under-motivated from a theoretical standpoint.**
>
>     The canvas expansion strategy is motivated by the observation that effective motion control typically requires pixel-aligned correspondences between trajectory points and object locations. To enable objects to move fully out of and back into the frame, we adjust the positional encoding—specifically, by extending RoPE and absolute position embeddings—so that the model can condition on an expanded spatial canvas beyond the original first frame.
>
>
>
> - **[W4-2]: Identity preservation relies on older techniques rather than recent advances (e.g., ConsisID)**
>
>     Our identity preservation method is inspired by Concat-ID (arXiv 2025), which is a more recent development than ConsisID (arXiv 2024). We found it particularly simple yet effective for maintaining identity consistency in our multi-stage framework.
>
>
> - **[W4-3]: There are few ablation studies on critical design choices like full-field loss or training stages.**
>
>     We provide detailed ablation results in Section D of the supplementary material, covering the effects of full-field loss, test-time video resolution, and positional encoding. Our multi-stage training strategy follows common practices in prior works, and we will clarify this motivation more explicitly in the final version.

---

> > ### Comment · Reviewer_XTCB · 2025-08-06
> >
> > Thanks for the authors' rebuttal. I would like to keep my positive rating.

---

### Official Review · Reviewer_MmwS · 2025-07-03

**Clarity:** 3
**Significance:** 3
**Originality:** 3
**Rating:** 4
**Confidence:** 4

**Summary:**

This paper introduces a novel video generation task dubbed “Frame In-N-Out,” which breaks through the limitations of first-frame boundaries to enable natural entry and exit of objects, flexible identity switching, and fine-grained control of motion trajectories within scenes. To support this, the authors present a new semi-automated dataset, a comprehensive evaluation protocol, and an efficient video diffusion Transformer architecture that ensures both identity preservation and controllable motion. Experimental results show that the proposed method substantially outperforms existing approaches in generation quality, controllability, and boundary-crossing capability, offering promising prospects for applications such as film production and advertising.

**Questions:**

- Is it possible to achieve some complex interactions? For example, can two people dance along specified trajectories and gradually exit the scene?
- If Frame In-N-Out is extended to real-world videos, especially those with complex lighting and physical constraints, what challenges might arise?
- How does the proposed model perform in terms of computational requirements and inference speed? How many frames can it support for video inference, and are there any optimization strategies?

**Ethical Concerns:**

["NO or VERY MINOR ethics concerns only"]

**Final Justification:**

The rebuttal addresses most of my concerns.

**Limitations:**

Yes

**Paper Formatting Concerns:**

No major formatting issues were found.

**Quality:**

3

**Strengths And Weaknesses:**

### Strengths
- This paper is the first to propose and formalize the novel video generation task of “Frame In-N-Out,” which effectively overcomes the limitation of traditional video generation being confined to in-frame objects, thereby expanding the application boundaries of video generation models.
- This paper proposes a novel video diffusion Transformer architecture that conditions on motion trajectories, reference subjects, and a boundary-free canvas, effectively enhancing the controllability of the generation process and the extensibility of the model.
- A high-quality Frame In-N-Out dataset with semi-automatic annotation support has been released, along with a dedicated evaluation system for the "in and out of frame" scenario to provide a reliable basis for model assessment.
- A high-quality Frame In-N-Out dataset was constructed, and a dedicated evaluation set tailored to this task was introduced, providing a solid foundation for model evaluation.
- The experiments are comprehensive, and the proposed method demonstrates advantages across multiple quantitative and subjective metrics, validating its effectiveness and superiority.
- Some physically consistent cases are impressive, such as picking up the cup shown in Appendix Fig. 2E; however, the hand movement appears somewhat stiff.

### Weakness
- The video scenes presented in the paper are relatively simple in terms of background, and there is a lack of scenarios involving crowded environments, frequent occlusions, significant background changes, or non-target motion interference.
- In the supplementary materials, some task scenes in the videos exhibit facial distortion (such as at 1 minute 40 seconds), which may reflect a general limitation of current video-generative models.
- The current generation paradigm is still primarily focused on 2D video, and has not yet addressed more complex scene composition issues in 3D space, such as motion in three-dimensional environments or virtual camera transitions.
- As mentioned in the limitations, this method is still unable to effectively understand the proportion between the given reference image and the background, and therefore cannot generate natural videos.

---

> ### Author Rebuttal · Authors · 2025-07-31
>
> Dear reviewer MmwS,
>
> This paper introduces the novel Frame In-N-Out video generation task, proposes a controllable diffusion Transformer architecture, releases a dedicated dataset and evaluation protocol, and demonstrates strong empirical results. We have prepared detailed response to your concerns.
>
> - **[W1]: The video scenes presented in the paper are relatively simple in terms of background.**
>
>     We appreciate the reviewer’s observation. The simplicity of the video background is a result of our data curation strategy, where we intentionally filter out scenes with complex background dynamics. Empirically, we found that dense point tracking remains challenging in such settings, even with state-of-the-art models like CoTrack3. To ensure high-quality training signals for Frame In-N-Out patterns, we apply a cycle tracking filter (Lines 126-136 in the main paper) to select reliable training samples. Our framework will continue to benefit from progress in dense tracking methods, and we plan to extend it to handle more complex scenes in future work.
>
>
> - **[W2]: Some videos exhibit facial distortion, which may reflect a general limitation of current video-generative models.**
>
>     Thank you for pointing this out! We fully agree that facial distortion remains a common limitation. Stronger base models and higher-quality training datasets may help alleviate this issue, and we plan to explore it in future work.
>
>
> - **[W3]: The current generation paradigm is still primarily focused on 2D video, and has not yet addressed more complex scene composition issues in 3D space.**
>
>     We appreciate this insightful comment. Full 3D object or camera motion control is indeed possible with our framework and was our initial goal. However, our early experiments revealed that curating high-quality videos with accurate 3D motion remains difficult. Even with strong models like CUT3R, estimates of camera rotation, translation, focal changes, and intrinsic camera parameters are not yet reliable enough on in-the-wild videos. For this reason, we focused first on 2D motion control as a foundational step, with full 3D controls as a longer-term objective.
>
>
> - **[W4]: As mentioned in the limitations, this method is unable to effectively understand the proportion between the given reference image and the background, and therefore cannot generate natural videos.**
>
>     We’d like to clarify that our limitation refers to the inherent ambiguity of 2D point-based conditioning, which is insufficient for precise 3D control such as object size, pose, and motion. That said, the generated results presented in Figure 6 of the supplementary material are reasonable outputs given the provided user input. We will make this clearer in the final version.
>
> ---
> Questions
> ---
>
> - **[Q1]: Is it possible to achieve some complex interactions? For example, can two people dance along specified trajectories and gradually exit the scene?**
>
>     Yes, our framework can support such complex interactions. For example, Figure 2(e) in the supplementary shows two balls interacting, and Figure 3(d) presents a flower swaying as two cats walk out from different directions. Our Stage 2 training incorporates identity references, making similar human interactions possible. While we’re limited by the text-only format in this rebuttal, we’d be happy to include more illustrative examples in the final version.
>
> - **[Q2]: If Frame In-N-Out is extended to real-world videos, especially those with complex lighting and physical constraints, what challenges might arise?**
>
>     There are several fundamental challenges. First, automatic text prompt generation during data curation remains difficult. Hallucinations and limited understanding of subtle visual details, such as lighting conditions or fine-grained motion, persist even with strong vision-language models like QWen2.5-VL-32B,. Second, adhering to physical constraints is a broad limitation for current video diffusion models. Many base models are pre-trained on datasets that include edited, CG, or even AIGC-generated content, often lacking real-world physical realism. The lack of transparency in these pretraining datasets further complicates efforts to correct such issues downstream. Overcoming these challenges may require more foundational improvements to the base models themselves.
>
>
> - **[Q7]: How does the proposed model perform in terms of computational requirements and inference speed? How many frames can it support for video inference, and are there any optimization strategies?**
>
>     Our model inherits the computational profile of the base model, CogVideoX, which was trained with a maximum resolution of 480×720 and up to 49 frames. For example, generating a 49-frame video at 448×640 resolution takes about 170 seconds on a single 80GB A100 GPU, with approximately 16 GB of memory usage. The model supports arbitrary resolutions below 480×720 and up to 49 frames per inference.  The training cost is 57GB for full finetune with 8-bit Adam. We did not apply additional optimization strategies and kept the inference procedure consistent with the base model. We will clarify these in the final version.

---

> > ### Comment · Reviewer_MmwS · 2025-08-04
> >
> > The rebuttal addresses most of my concerns.

---

### Decision · Program_Chairs · 2025-09-17

**Decision:**

Accept (poster)

**Comment:**

This paper receives positive scores. The reviewers raise some concerns regarding limited technical novelty, simple video scenes in terms of background, the lack of video scenarios with complex interactions and camera motion, computational requirements and inference speed, resolution mismatch between training and inference, inferior generated videos with facial distortion, evaluation constraints, etc. Most of the questions are addressed in the rebuttal and recognized by reviewers. Eventually, this paper is recommended to be accepted. Authors should incorporate these review feedbacks, and polish the paper writing in the final version.